# Digestibility of dinosaur food plants revisited and expanded: Previous data, new taxa, microbe donors, foliage maturity, and seasonality

**Mariah M. Howell** [1]*, **Carole T. Gee**[1], **Christian Böttger**[2], **Karl-Heinz Südekum**[2]

**1** Division of Paleontology, Institute of Geosciences, University of Bonn, Bonn, Germany, **2** Institute of Animal Science, University of Bonn, Bonn, Germany

* mariah.howell@uni-bonn.de

**Data Availability Statement:** All relevant data are within the paper and its Supporting Information files.

## Abstract

Although the living relatives of the Mesozoic flora were once assumed to constitute a nutritionally poor diet for dinosaur herbivores, *in vitro* fermentation of their foliage has shown that gymnosperms, ferns, and fern relatives can be as highly digestible as angiosperm grasses and dicots. Because nutritional information cannot be preserved in the fossil record, this laboratory approach, first published in 2008, provides a novel alternative to evaluate the digestive quality of the plants that were available to dinosaur megaherbivores such as sauropods. However, very few further studies have since been conducted to supplement and confirm the high fermentative capacity of nonangiospermous taxa. Here we show that the living relatives of the Araucariaceae and Equistaceae are consistently highly digestible, even between taxa and when influenced by environmental and biological factors, while fern taxa are inconsistent on the family level. These results reinforce previous findings about the high energetic potential of Jurassic-age plant families. Fourteen species of fern and gymnosperm foliage from five Jurassic families were collected in the spring and fall, then analyzed for their digestibility using the *in vitro* Hohenheim gas test. *Equisetum*, *Araucaria*, and *Angiopteris* were the most digestible genera in both seasons, while *Agathis*, *Wollemia*, and *Marattia* were the least digestible. The season in which specimens were collected was found to have to a significant effect on gas production in four out of 16 samples ($P < 0.05$). Furthermore, leaf maturity influences digestibility in *Marattia attenuata* ($P < 0.05$), yet not in *Cyathea cooperi* ($P = 0.24$). Finally, the species of the rumen fluid donor did not influence digestibility ($P = 0.74$). With the original data set supplemented by one new genus and four species, this study confirms and expands previous results about the nutritional capacity of the living relatives of the Jurassic flora.

**Funding:** The authors received no specific funding for this work.

**Competing interests:** The authors have declared that no competing interests exist.

## Introduction

Although innovative research in 2008 refuted the long-held hypothesis that nonangiospermous plants are nutritionally poor compared to flowering plants by demonstrating that horsetails, ferns, and conifers have the capacity to be as digestible as grasses [1], little additional research has been conducted to confirm and further understand the digestive properties of Jurassic-age plant families. The flora of this geological period predominantly consisted of ferns, horsetails, and gymnosperms such as seed ferns, cycads, bennettititaleans, ginkgoes, and conifers. Among the conifers, the Araucariaceae, Podocarpaceae, and Cheirolepidiaceae flourished globally [2]. Though they dominate modern ecosystems and herbivorous diets, angiosperms did not appear until the Early Cretaceous, [3]; most of the phylogenetic and ecological radiation of the angiosperms occurred in the Late Cretaceous [4].

As such, the diets of herbivorous dinosaurs, including sauropods, would have been comprised of gymnosperms, ferns, and horsetails for much of the Mesozoic. Determining the digestive properties of the Jurassic flora would therefore be key to understanding the feeding ecology of these megaherbivores. However, this information cannot be preserved through fossilization, and for many years, these nonangiospermous taxa were suspected to be poorly digestible compared to flowering plants. The reconstruction of sauropod diets was thus driven by the question of how such a nutritionally poor diet influenced the evolution and biology of the largest land animals to ever live [5–8].

In the pivotal study of Hummel et al. [1], they performed a series of *in vitro* digestive experiments with the Hohenheim gas test to examine the digestibility and energetic yield of the living relatives of the Mesozoic flora. The Hohenheim gas test is a well-known, standardized laboratory method that analyzes the digestibility of livestock feed by fermenting plant matter in the rumen liquid of cows, sheep, or other ruminants. This method allows the ruminal microflora to break down the digestible fiber in the plant matter in the lab, producing gas [9, 10]. The amount of gas produced directly correlates to the digestibility of the plant matter and, therefore, the amount of metabolizable energy that an herbivore can extract from any given plant [11]. This approach contrasts to the ash test that was used by Weaver [5], which quantified the caloric value of both the digestible and indigestible fractions [1]. With the ash test, it was concluded that cycads were the best source of energy for sauropod dinosaurs, while *Equisetum* was the worst [5].

To approximate the digestibility of ancient plants by applying the Hohenheim gas test to the nearest living relatives of Jurassic plants, it is possible to compare the digestibility and relative amount of energy that dinosaurs may have been able to extract from the nonangiospermous flora of the Mesozoic. While not all lineages—for instance, the Bennettitales, Cheirolepidiaceae, or Pteridospermales (seed ferns)—have living members that can be tested with this method, a number of Jurassic families have survived until today. These include the Equisetaceae, Araucariaceae, Cyatheaceae, Marattiaceae, and Osmundaceae.

The results of Hummel et al. [1] showed that many of these living relatives, particularly those of the Equisetaceae and Araucariaceae, equaled or even surpassed angiosperm grasses and dicot leaves in digestibility and energy yield. Other species, such as ferns in the Marattiaceae and Osmundaceae, were also nearly as digestible as grasses, while *Ginkgo* was shown to be as digestible as dicot leaf browse while also providing as much crude protein as grasses. Later, Gill et al. [12] used the Hohenheim gas test and found similar, or even greater, digestibility in plants grown under experimentally increased $CO_2$ levels that mimicked Mesozoic atmospheric conditions, which had been hypothesized to decrease the nutritional quality of the Mesozoic flora [7, 8]. Concrete data on the nutrition and digestibility of Jurassic plant families also allowed for estimations of daily food intake tailored to different dietary compositions [1, 12].

However, despite the new approach of this method, there has been no additional research to substantiate the findings and follow up with a larger survey of taxa.

The present study quantifies the digestibility of the nearest living relatives of five Jurassic-age families using the Hohenheim gas test, confirming the data resulting from previous research and expanding the data set with additional species and analyses. For example, *Wollemia*, the third genus in the Araucariaceae, is analyzed here and completes the data set on the family, while two previously unanalyzed species of the genus *Agathis* increases the number of *Agathis* species tested from one to three. Among the ferns, the existing data on the Marattiaceae, which was previously represented by only *Angiopteris evecta*, was also expanded with the inclusion of a second genus, *Marattia*. Furthermore, multiple factors that affect digestibility, such as microbe donor species, foliage maturity, and seasonality, are also analyzed.

## Materials and methods

### Sample collection

Foliage from 14 species representing the nearest living relatives of Mesozoic flora from the families of the Equisetaceae (2 species), Marattiaceae (2 species), Osmundaceae (1 species), Cyatheaceae (1 species), and Araucariaceae (8 species), as well as the marine angiosperm family of eelgrasses, Zosteraceae (1 species) (Table 1), was collected. In the case of the Equisetaceae, above-ground green shoots were harvested, while leaf-bearing twigs were collected from the Araucariaceae. Individual fronds were taken from the Marattiaceae, Osmundaceae, and Cyatheaceae. In two species, *Marattia attenuata* and *Cyathea cooperi*, a distinction was made between "older" and "younger" leaves. All samples were clipped from the plant using garden shears, except for two clumps of *Zostera marina* leaves that were picked up during low tide.

All specimens were collected from the Huntington Library, Art Collections, and Botanical Gardens in San Marino, California, USA, except for *Zostera marina*, which was collected 50 m north of La Jolla Pier on the southern coast of California. Sampling initially occurred in April 2018 (hereafter called "spring"), then a second collection was gathered from the same individual plants in September 2018 ("fall"). *Zostera marina* was collected in spring but was not collected again in the fall. Specimens were dried for 48 hours at 60°C as soon as possible after collection, following the protocol established by Menke and Steingass (1988). Four specimens had not dried completely and were thus heated at 60°C for an additional three hours (spec. nos. 18-28-3; 18-28-4; 18-28-7; 18-28-8). The dried samples were then milled successively using 3 mm and 1 mm sieves using a Retsch SM1 cutting mill (Haan, Germany).

### Gas production

The digestibility of each species was determined using a modified version of the *in vitro* Hohenheim gas test [9, 10] at the Institute of Animal Science, University of Bonn, Germany. Rumen fluid was collected from either ruminally cannulated cattle (steer; trials 1 and 2) or ruminally cannulated sheep (trials 3 and 4) fed a standardized diet (hay and concentrate; 70:30) according to the animal maintenance requirements. The animals used for obtaining liquid ruminal content were kept in strict accordance with the German Animal Welfare legislation. All experimental procedures were approved in advance by the Animal Care Committee of the state of North Rhine-Westphalia in Germany (File Number 84–02.04.2017.A247 [cattle] and 81–02.05.40.18.008 [sheep]).

An nXP buffer solution (6 g $NH_4HCO_3$ + 33 g $NaHCO_3$ per liter) was prepared according to the mixture outlined by Menke and Steingass [10] but modified to the concentration recommendations of Liu et al. [13] to accommodate the long incubation period. The milled plant material (200 mg ± 10 mg), rumen fluid containing fermentative microbes, and buffer solution

**Table 1. Specimens collected and evaluated using the Hohenheim gas test.**

| Collection Date | Spec. No. | Species | Family | Notes |
|---|---|---|---|---|
| 4/7/2018 | 18-28-1 | *Zostera marina* | Zosteraceae | 2 clumps on beach, low tide |
| 4/9/2018 | 18-28-3 | *Angiopteris evecta* | Marattiaceae | Frond 1 |
| 4/9/2018 | 18-28-4 | *Angiopteris evecta* | Marattiaceae | Frond 2 |
| 4/9/2018 | 18-28-5 | *Equisetum giganteum* | Equisetaceae | |
| 4/9/2018 | 18-28-6 | *Osmunda regalis* | Osmundaceae | Access. no. 87695*1 |
| 4/9/2018 | 18-28-7 | *Marattia attenuata* | Marattiaceae | Young leaf |
| 4/9/2018 | 18-28-8 | *Marattia attenuata* | Marattiaceae | Older leaf, same individual |
| 4/9/2018 | 18-28-9 | *Equisetum hyemale* | Equisetaceae | |
| 4/9/2018 | 18-28-10 | *Cyathea cooperi* | Cyatheaceae | Older leaf |
| 4/9/2018 | 18-28-11 | *Cyathea cooperi* | Cyatheaceae | Younger leaf, same individual |
| 4/9/2018 | 18-28-12 | *Araucaria laubenfelsii* | Araucariaceae | |
| 4/9/2018 | 18-28-13 | *Araucaria columnaris* | Araucariaceae | Access. no. 33438 |
| 4/9/2018 | 18-28-14 | *Araucaria heterophylla "glauca"* | Araucariaceae | Access. no. 32460 |
| 4/9/2018 | 18-28-15 | *Araucaria bidwillii* | Araucariaceae | Access. no. 87754 |
| 4/9/2018 | 18-28-16 | *Agathis robusta* | Araucariaceae | Access. no. 18450 |
| 4/9/2018 | 18-28-17 | *Agathis lanceolata* | Araucariaceae | Access. no. 71561 |
| 4/9/2018 | 18-28-18 | *Agathis australis* | Araucariaceae | Access. no. 71261 |
| 4/9/2018 | 18-28-19 | *Wollemia nobilis* | Araucariaceae | Access. no. 95020*10 |
| 4/9/2018 | 18-28-20 | *Wollemia nobilis* | Araucariaceae | Access. no. 95020*9 |
| 9/10/2018 | 18-28-21 | *Angiopteris evecta* | Marattiaceae | Frond 1, low light |
| 9/10/2018 | 18-28-22 | *Angiopteris evecta* | Marattiaceae | Frond 2, direct light |
| 9/10/2018 | 18-28-23 | *Equisetum giganteum* | Equisetaceae | |
| 9/10/2018 | 18-28-24 | *Osmunda regalis* | Osmundaceae | Access no. 87695*1 |
| 9/10/2018 | 18-28-25 | *Marattia attenuata* | Marattiaceae | Young leaf |
| 9/10/2018 | 18-28-26 | *Marattia attenuata* | Marattiaceae | Older leaf, same individual |
| 9/10/2018 | 18-28-27 | *Equisetum hyemale* | Equisetaceae | |
| 9/10/2018 | 18-28-28 | *Cyathea cooperi* | Cyatheaceae | Older leaf |
| 9/10/2018 | 18-28-29 | *Cyathea cooperi* | Cyatheaceae | Younger leaf, same individual |
| 9/10/2018 | 18-28-30 | *Araucaria laubenfelsii* | Araucariaceae | |
| 9/10/2018 | 18-28-31 | *Araucaria columnaris* | Araucariaceae | Access. no. 33438 |
| 9/10/2018 | 18-28-32 | *Araucaria heterophylla "glauca"* | Araucariaceae | Access. no. 32460 |
| 9/10/2018 | 18-28-33 | *Araucaria bidwillii* | Araucariaceae | Access. no. 87754 |
| 9/10/2018 | 18-28-34 | *Agathis robusta* | Araucariaceae | Access. no. 18450 |
| 9/10/2018 | 18-28-35 | *Agathis lanceolata* | Araucariaceae | Access. no. 71561 |
| 9/10/2018 | 18-28-36 | *Agathis australis* | Araucariaceae | Access. no. 71261 |
| 9/10/2018 | 18-28-37 | *Wollemia nobilis* | Araucariaceae | Access. no. 95020*10 |
| 9/10/2018 | 18-28-38 | *Wollemia nobilis* | Araucariaceae | Access. no. 95020*9 |

were loaded into sealed syringes, then incubated in a rotating syringe rack at 39˚C for 96 hours to allow for maximum potential digestion, thereby removing time as a limiting factor. Gas volume, in milliliters, was recorded at the 4, 8, 12, 24, 32, 48, 56, 72, and 96-hour marks and subsequently corrected for blank gas production and expressed as ml/200 mg plant dry matter (DM).

Four total replicates of each sample were incubated in four runs on different weeks, except for the spring *Osmunda regalis* (spec. no. 18-28-6) and young spring *Marattia attenuata* (spec. no. 18-28-7), which were each tested in two runs due to low sample volume. One sample, the fall leaves of *Araucaria heterophylla* "*glauca*" (spec. no. 18-26-33), leaked during the third run and was retested with two replicates, for a total of five complete data sets. Three blanks

containing rumen fluid–buffer solution without samples, three hay standard samples, and three concentrate standard samples were included in each run for the calibration of results. The hay standard and concentrate samples were obtained from the Department of Animal Nutrition at Hohenheim University in Stuttgart, Germany.

## Statistical analyses

The parameters for gas production were fitted by a non-linear regression using PROC NLIN in SAS [14], and the following model was used to plot the kinetics of gas production in Excel 2016 [15]:

$$Gp = a + b(1 - e^{-ct})$$

wherein *Gp* represents total gas production (ml/200 mg DM), *a* + *b* is the maximum gas production (ml/200 mg DM), *a* is the initial gas production of the rapidly digestible fraction (ml/200 mg DM), *b* is the gas production of the remaining slowly digestible fraction (ml/200 mg DM), *c* is the rate of gas production (h$^{-1}$), and *t* is time (h) [16, 17]. Differences between net gas production between species, family, season, and maturity were calculated using ANOVAs and Tukey post-hoc tests using R version 4.2.2 [18] with a significance threshold of 0.05 (cf. [19]). Results were plotted using Excel 2016 [15].

## Repository

The milled plant material from this experiment is archived in the collection of the Institute of Animal Science, at the University of Bonn, Germany.

## Results

Gas production varied significantly between species (*P* < 0.05) as well as between families (*P* < 0.05), genera (e.g., Araucariaceae, *P* < 0.05) and sometimes even within species (e.g., *Marattia attenuata*, *P* < 0.05). The Equisetaceae were by far the most fermentatively productive family, followed by the Osmundaceae, Araucariaceae, Cyatheaceae, and finally the Marattiaceae, with each of the latter families producing similar, slowly-building kinetic curves, in contrast to the rapid and high initial production in the Equisetaceae which is then followed by a plateau (Fig 1).

On the species level, *Equisetum hyemale* was found to be the most digestible species in both seasons (mean: 58.33 ml/200 mg DM; standard deviation: 3.45), while the mature leaves of *Marattia attenuata* were the least digestible in both collection periods (mean: 9.52 ml/200 mg; standard deviation: 3.12; Fig 2A and 2B). In only four of 16 samples is there a significant change (*Araucaria bidwillii*, *P* = 0.003; *Osmunda regalis*, *P* = 0.001; older *Marattia attenuata*, *P* < 0.05; younger *Cyathea cooperi*, *P* < 0.05) in gas production between the two collection periods (Tables 2 and 3).

In all but three specimens, those collected in the spring yielded comparatively more gas than the same individuals did in the fall collection (Fig 3).

Ferns and fern allies were variable (*P* = 0.73) in their net gas production, and only *Osmunda regalis* had significantly more gas output in the spring (*P* = 0.001). Rate of gas production (*c*) of both *Equisetum* species and *Angiopteris evecta* was comparatively high in both seasons (Tables 2 and 3), with rapid initial production that leveled off soon after the first 24 hours; at least 79% of gas production took place in the first 24 hours, with *A. evecta* reaching over 90% of gas production during that time (Fig 4), making it the most rapidly digestible of any species in both seasons (Tables 2 and 3). Conversely, the leaves of *Marattia attenuata* had low rates of

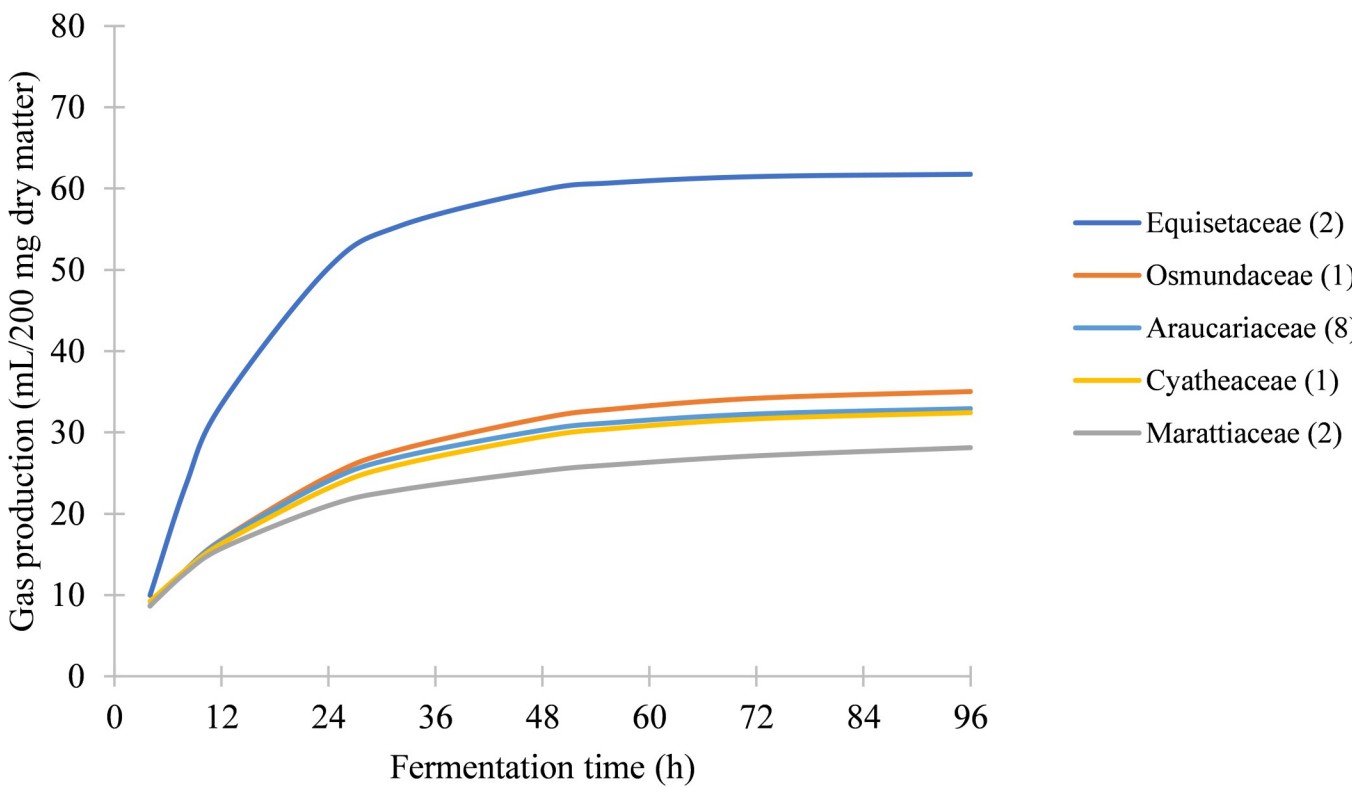

**Fig 1. Mean gas production by family for both seasons.** The quantities in parentheses indicate the number of species tested in each family.

gas production and achieved 60% or less of its 96-hour production within the first 24 hours (Tables 2 and 3 and Fig 4).

In terms of absolute maximum gas production, both species of *Equisetum* outperformed all other species, with the spring samples of *E. hyemale* overtaking *E. giganteum* immediately, but the fall samples only after a lead of twelve hours. In the spring, *Osmunda regalis* was the most digestible fern despite its slower rate of production (mean: 30.75 ml; standard deviation: 1.63), but in the fall, *Angiopteris evecta* (mean: 33.28 ml; standard deviation: 3.93) and both samples of *Cyathea cooperi* were more digestible. *Marattia attenuata* was the least digestible fern, regardless of leaf maturity or season (mean: 1.1; standard deviation: 0.54; Fig 2A and 2B). However, leaf maturity still significantly affected digestibility in *Marattia attenuata* (Fig 5A; $P < 0.05$) but did not affect the digestibility of *Cyathea cooperi* (Fig 5B; $P = 0.24$).

Of the eight ferns and fern allies tested, only *Osmunda regalis* was influenced by seasonality (Fig 3). Within the same individual organism of *Marattia attenuata*, the net gas production of younger leaves was not significantly influenced by seasonality ($P = 0.93$), but the production of older leaves was influenced ($P < 0.05$). This was reversed in *Cyathea cooperi*, in which neither the younger leaves experienced significant seasonal alterations in digestibility ($P < 0.05$) but not the older leaves ($P = 0.06$). *Osmunda regalis* experienced the greatest seasonal shift in gas production volume of any species, producing 44% more gas in the spring than the fall (Tables 2 and 3 and Fig 3).

In general, the Araucariaceae had lower rates of gas production than many of the ferns but achieved comparable maximum gas production near the end of the experiment (Tables 2 and 3). Members of this family tended towards slower production curves and did not begin to plateau until between 72 and 96 hours (Fig 6A). For example, in both seasons, the most digestible

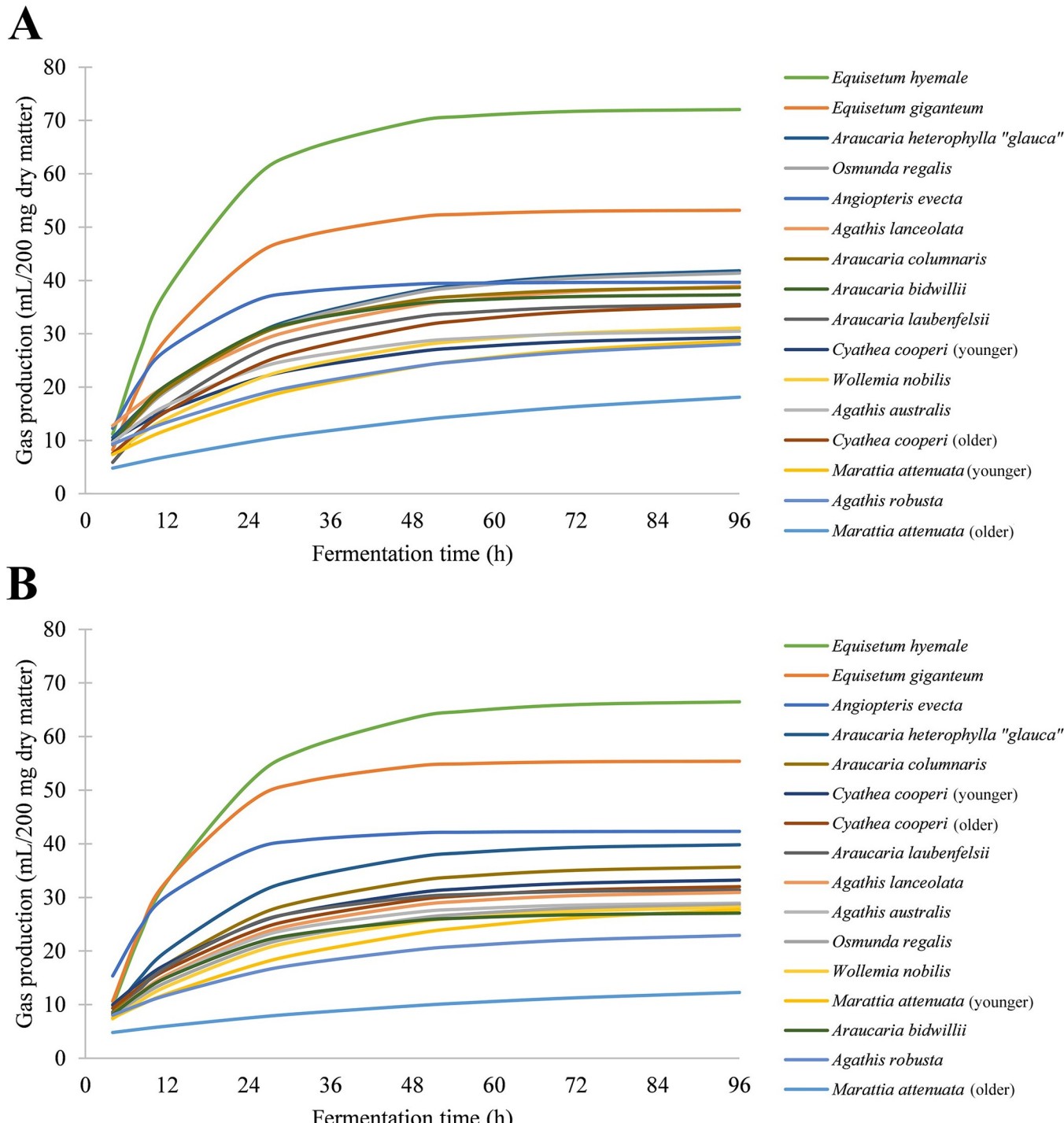

**Fig 2. Gas production kinetics by season.** (A) Cumulative gas production of species collected in the spring (April 2018). (B) Cumulative gas production of species collected in the fall (September 2018).

araucarian species was *Araucaria heterophylla* "*glauca*," which produced similar gas volumes as the ferns *Osmunda regalis* and *Angiopteris evecta* after 56 hours of digestion, despite its much slower initial production and more gradual digestion curve (Fig 2A and 2B). Of the

**Table 2. Parameters for the fermentation of samples collected in the spring (April 2018).**

| Species | Cumulative gas production (ml/200 mg dry matter) | | | | | | | | | | |
| --- | --- | --- | --- | --- | --- | --- | --- | --- | --- | --- | --- |
| | 4 h | 8 h | 12 h | 24 h | 32 h | 48 h | 56 h | 72 h | 96 h | *a + b* | *c* |
| *Angiopteris evecta* | 13.3 | 22.4 | 28.6 | 37.7 | 40.0 | 41.6 | 41.8 | 42.0 | 42.0 | 42.0 | 0.095 |
| *Angiopteris evecta* | 11.2 | 19.8 | 25.5 | 33.7 | 35.7 | 37.0 | 37.2 | 37.3 | 37.3 | 37.3 | 0.099 |
| *Equisetum giganteum* | 8.2 | 20.4 | 29.2 | 43.9 | 48.2 | 51.8 | 52.4 | 53.0 | 53.1 | 53.2 | 0.079 |
| *Osmunda regalis* | 9.3 | 14.8 | 19.3 | 28.9 | 32.9 | 37.5 | 38.9 | 40.4 | 41.4 | 41.9 | 0.056 |
| *Marattia attenuata* (younger) | 7.4 | 9.8 | 12.0 | 17.2 | 19.9 | 23.7 | 25.1 | 27.0 | 28.7 | 30.5 | 0.038 |
| *Marattia attenuata* (older) | 4.8 | 5.9 | 7.0 | 9.7 | 11.2 | 13.7 | 14.7 | 16.4 | 18.1 | 21.6 | 0.017 |
| *Equisetum hyemale* | 11.2 | 26.7 | 38.3 | 58.1 | 64.3 | 69.7 | 70.8 | 71.7 | 72.0 | 72.1 | 0.074 |
| *Cyathea cooperi* (younger) | 10.0 | 13.0 | 15.6 | 21.1 | 23.6 | 26.6 | 27.5 | 28.6 | 29.3 | 29.8 | 0.041 |
| *Cyathea cooperi* (older) | 7.6 | 11.8 | 15.5 | 23.4 | 26.9 | 31.7 | 32.6 | 34.2 | 35.3 | 35.9 | 0.041 |
| *Araucaria laubenfelsii* | 5.9 | 11.7 | 16.5 | 25.7 | 29.3 | 33.0 | 34.0 | 35.0 | 35.5 | 35.7 | 0.055 |
| *Araucaria columnaris* | 9.2 | 15.0 | 19.7 | 28.9 | 32.5 | 36.2 | 37.2 | 38.2 | 38.7 | 38.9 | 0.055 |
| *Araucaria heterophylla* "glauca" | 10.6 | 15.7 | 20.1 | 29.3 | 33.2 | 37.8 | 39.2 | 40.8 | 41.8 | 42.3 | 0.045 |
| *Araucaria bidwillii* | 9.6 | 15.7 | 20.5 | 29.3 | 32.5 | 35.6 | 36.3 | 37.0 | 37.3 | 37.4 | 0.062 |
| *Agathis robusta* | 9.3 | 11.5 | 13.4 | 18.1 | 20.5 | 23.8 | 25.0 | 26.7 | 28.1 | 29.5 | 0.029 |
| *Agathis lanceolata* | 12.7 | 16.8 | 20.3 | 27.8 | 31.1 | 35.2 | 36.4 | 37.9 | 38.9 | 39.5 | 0.041 |
| *Agathis australis* | 9.7 | 13.7 | 16.6 | 23.0 | 25.5 | 28.4 | 29.1 | 30.0 | 30.5 | 30.7 | 0.050 |
| *Wollemia nobilis* | 7.5 | 10.7 | 13.5 | 19.6 | 22.2 | 25.5 | 26.5 | 27.7 | 28.5 | 29.0 | 0.042 |
| *Wollemia nobilis* | 7.3 | 11.3 | 14.7 | 22.2 | 25.6 | 29.7 | 31.0 | 32.6 | 33.7 | 34.3 | 0.040 |

For each sample, cumulative gas production (ml/200 mg dry matter) up to 96 h and fermentative parameters, in which *a + b* is maximum gas production (ml/200 mg dry matter) and *c* is rate of gas production ($h^{-1}$), are given.

**Table 3. Parameters for the fermentation of samples collected in the fall (September 2018).**

| Species | Cumulative gas production (ml/200 mg dry matter) | | | | | | | | | | |
| --- | --- | --- | --- | --- | --- | --- | --- | --- | --- | --- | --- |
| | 4 h | 8 h | 12 h | 24 h | 32 h | 48 h | 56 h | 72 h | 96 h | *a + b* | *c* |
| *Angiopteris evecta* | 14.1 | 23.0 | 28.7 | 36.1 | 37.7 | 38.6 | 38.7 | 38.7 | 38.8 | 38.8 | 0.112 |
| *Angiopteris evecta* | 16.6 | 25.6 | 31.9 | 41.3 | 43.7 | 45.4 | 45.6 | 45.8 | 45.8 | 45.9 | 0.093 |
| *Equisetum giganteum* | 10.6 | 23.8 | 33.2 | 47.6 | 51.5 | 54.5 | 54.9 | 55.3 | 55.4 | 55.42 | 0.087 |
| *Osmunda regalis* | 8.0 | 11.4 | 14.3 | 20.3 | 22.9 | 26.0 | 26.9 | 28.0 | 28.7 | 29.1 | 0.044 |
| *Marattia attenuata* (younger) | 7.5 | 9.9 | 12.0 | 17.1 | 19.6 | 23.2 | 24.4 | 26.2 | 27.7 | 29.2 | 0.029 |
| *Marattia attenuata* (older) | 4.8 | 5.4 | 6.0 | 7.6 | 8.4 | 9.8 | 10.4 | 11.3 | 12.3 | 14.23 | 0.017 |
| *Equisetum hyemale* | 9.9 | 22.9 | 33.0 | 51.3 | 57.6 | 63.5 | 64.8 | 66.0 | 66.5 | 66.6 | 0.055 |
| *Cyathea cooperi* (younger) | 10.0 | 14.2 | 17.6 | 24.7 | 27.5 | 30.8 | 31.6 | 32.6 | 33.2 | 33.5 | 0.047 |
| *Cyathea cooperi* (older) | 9.3 | 13.3 | 16.6 | 23.4 | 26.2 | 29.4 | 30.3 | 31.4 | 32.0 | 32.3 | 0.049 |
| *Araucaria laubenfelsii* | 7.5 | 12.9 | 17.1 | 24.7 | 27.4 | 30.0 | 30.6 | 31.1 | 31.4 | 31.4 | 0.064 |
| *Araucaria columnaris* | 8.4 | 13.5 | 17.6 | 26.0 | 29.3 | 33.0 | 34.0 | 35.1 | 35.7 | 35.9 | 0.051 |
| *Araucaria heterophylla* "glauca" | 8.6 | 15.0 | 20.1 | 30.0 | 33.6 | 37.4 | 38.4 | 39.3 | 39.8 | 40.00 | 0.057 |
| *Araucaria bidwillii* | 8.2 | 12.0 | 15.1 | 21.1 | 23.3 | 25.6 | 26.2 | 26.8 | 27.1 | 27.7 | 0.057 |
| *Agathis robusta* | 8.0 | 10.0 | 11.8 | 15.8 | 17.7 | 20.2 | 21.0 | 22.1 | 22.9 | 23.6 | 0.034 |
| *Agathis lanceolata* | 8.2 | 12.2 | 15.6 | 22.4 | 25.2 | 28.4 | 29.3 | 30.3 | 30.9 | 31.2 | 0.048 |
| *Agathis australis* | 7.5 | 11.8 | 15.2 | 21.9 | 24.5 | 27.2 | 27.9 | 28.6 | 28.9 | 29.1 | 0.055 |
| *Wollemia nobilis* | 7.6 | 11.1 | 14.0 | 20.2 | 22.9 | 26.0 | 27.0 | 28.1 | 28.8 | 29.2 | 0.044 |
| *Wollemia nobilis* | 7.1 | 10.3 | 13.0 | 18.8 | 21.4 | 24.6 | 25.6 | 26.8 | 27.6 | 28.1 | 0.041 |

For each sample, cumulative gas production (ml/200 mg dry matter) up to 96 h and fermentative parameters, in which *a + b* is maximum gas production (ml/200 mg dry matter) and *c* is rate of gas production ($h^{-1}$), are given.

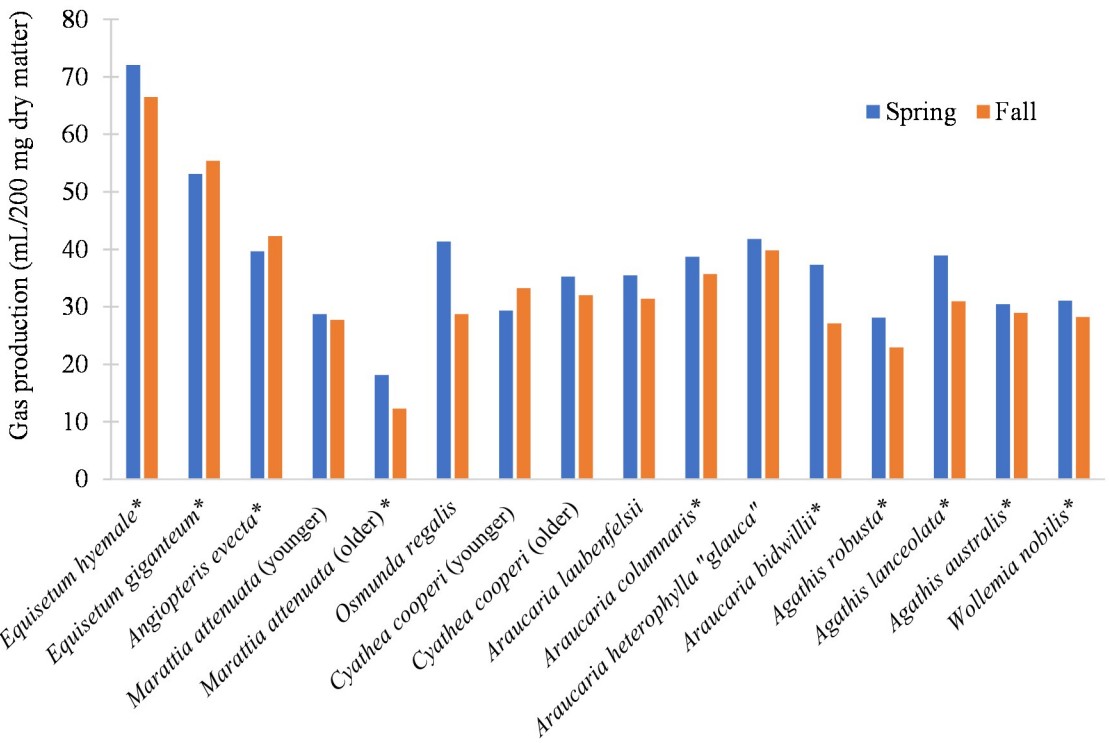

**Fig 3. Mean gas production by season.** Blue bars represent the maximum gas production ($a + b$) of samples collected during the spring (April 2018). Red bars the maximum gas production ($a + b$) of samples collected in the fall (September 2018). Asterisk indicates $P < 0.05$ for seasonal affects within the species.

three genera in the family, the most productive genus was *Araucaria*, while *Agathis* and *Wollemia* achieved similar gas production curves (Fig 6B). *Araucaria* was significantly more productive than either *Wollemia* or *Agathis* ($P < 0.05$; $P < 0.05$), but the latter genera did not significantly differ from each other ($P = 0.88$). However, generic affinity was not necessarily a predictor of digestibility; in the spring, *Agathis lanceolata* was the second most digestible species in the family, while *Agathis robusta* was the least digestible (Fig 6A). *Agathis robusta* was also particularly slow to ferment, with less than 60% of its gas production occurring within the first 24 hours (Fig 4) and the lowest rate of production of any araucarian conifer in either season (Tables 2 and 3). Among the Araucariaceae, only *Araucaria bidwillii* was found to be significantly affected by seasonal changes (Fig 3).

The marine angiosperm *Zostera marina*, known as common eelgrass, is excluded here from data interpretation because its cumulative gas production was too poor to be fitted to the mathematical model. After 24 hours, eelgrass produced only 4.1 ml/200 mg DM of gas, and only 10.2 ml/200 mg DM of gas after 96 hours. Additionally, the rumen fluid donor species did not have a significant influence on maximum gas production (P = 0.74), particularly after the first 24 hours of production.

## Discussion

Among those tested here, a select few living relatives of the Mesozoic flora display a high capacity for digestibility and energetic yield. In particular, the Equisetaceae and the Araucariaceae consistently yield high quantities of available energy, indicating a high potential for this

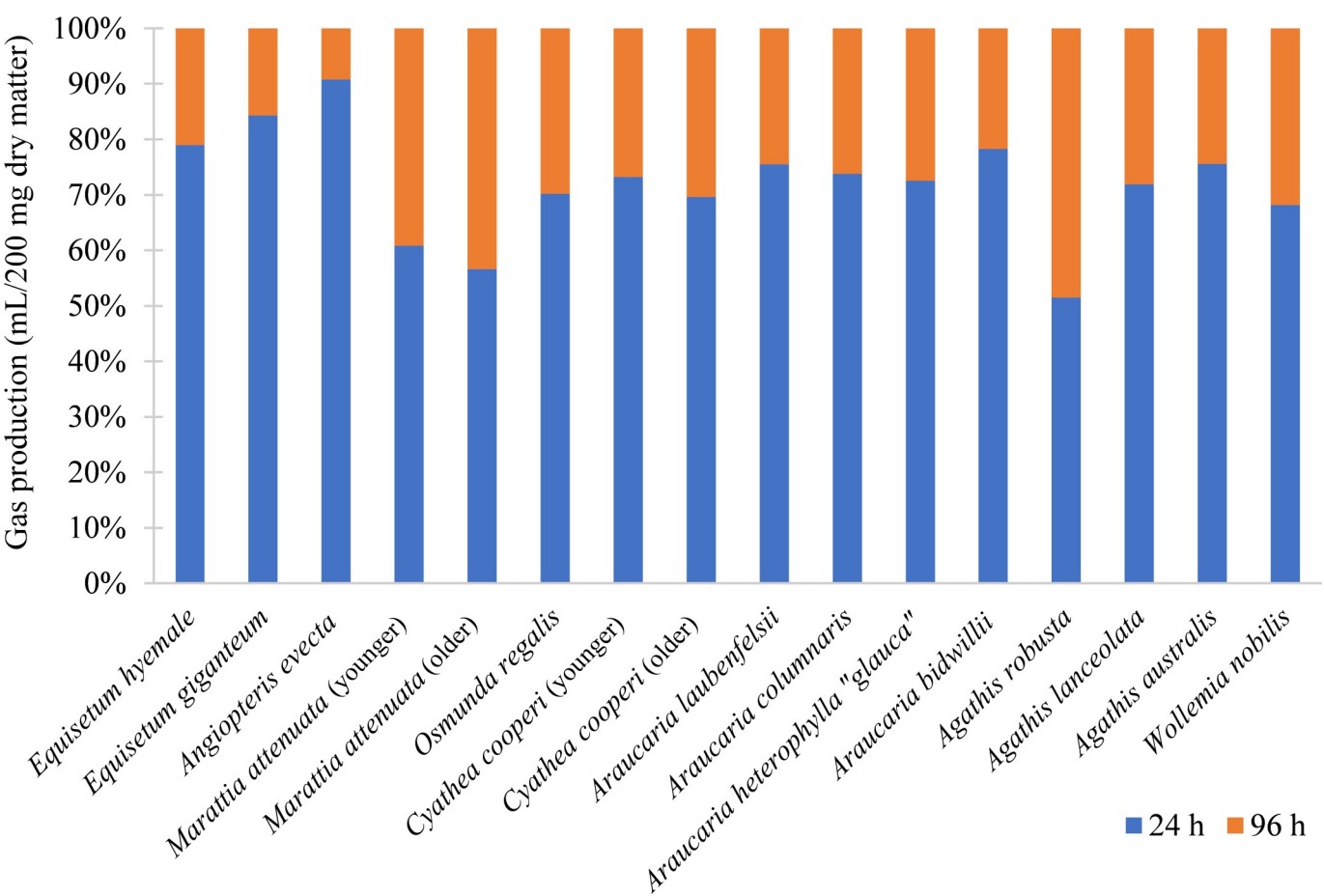

**Fig 4. Digestion time.** Percentage of maximum gas production achieved within the first 24 hours (blue) and the subsequent 72 hours (orange), for a total of 96 hours of digestion.

family to have been a targeted food source of herbivorous dinosaurs, if the nutritional qualities of the ancient plants were similar to those of their living descendants. Although chemical information of this type is not preserved, the structural morphology of *Equisetum*, for example, has remained largely unchanged since the Middle Triassic [20], suggesting that the digestive properties may also have been conserved through time.

The other families included in this experiment show greater digestive variation between species, making family-level generalizations of their dietary usefulness difficult. While the family Marattiaceae contains the fern species yielding the lowest energy, *Marattia attenuata*, it also has the fern species with the highest energy yield, *Angiopteris evecta* (Fig 2A and 2B). These results are consistent with, and even exceed, those previously found by Hummel et al. [1].

### Digestibility of *Equisetum*

*Equisetum*, as previously shown by Hummel et al. [1] and Gill et al. [12], continues to outperform all other groups in energy yield. In the current study, *E. hyemale* had the highest 96-hour gas production of any species, with a spring volume of 72.0 ml/200 mg DM (Table 2). The maximum gas production values for this species are marginally exceeded only by *E. hyemale* grown by Gill et al. [12] in 1200 ppm atmospheric $CO_2$ conditions, which reached 72.4 ml/200 mg DM. However, Gill et al. [12] determined that *E. hyemale* was not significantly affected by

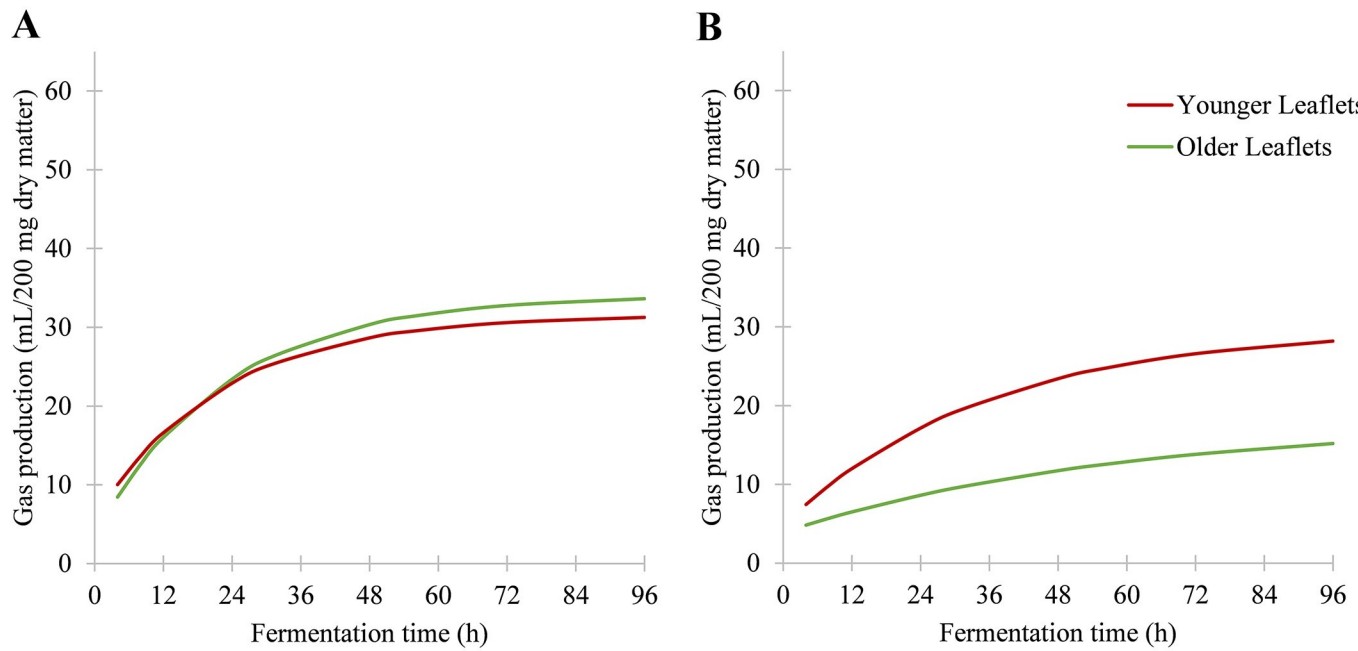

**Fig 5. Influence of leaf maturity on the fermentation of two species.** (A) Fermentative behavior of younger and older leaflets of *Cyathea cooperi* ($P = 0.24$). (B) Fermentative behavior of younger and older leaflets of *Marattia attenuata* ($P < 0.05$).

$CO_2$ concentration, which is further supported by our samples which grew outdoors in ambient $CO_2$ conditions yet achieved markedly similar digestibility. The consistency of these findings may also suggest that our results approach the upper digestibility potential of *E. hyemale*. Compared to the other species in this genus, *E. hyemale* was found to produce significantly more gas than *E. giganteum* ($P < 0.05$), which itself produced significantly more gas than all other species included in the current study ($P < 0.05$).

Although the high silica content of *Equisetum* [cf. 21] is thought to inhibit the digestibility of foliage [22], the overall high performance of this genus reiterates it would have been a consistently nutritious option for herbivorous dinosaurs. While both *E. hyemale* and *E. giganteum* exhibit a reduction in digestibility in the fall, they are still more digestible than any other sample (Fig 3), far exceeding the standards for both modern angiosperm grasses and dicot leaves determined by Hummel et al. [23]. Furthermore, the cosmopolitan distribution of *Equisetum* and its relatives during the Jurassic and Cretaceous [24, 25] would have made it commonly available to the global fauna of herbivorous dinosaurs in the Mesozoic [26].

### Digestibility of ferns

Among the ferns, digestibility is more inconsistent as they tend to be affected by season and leaf maturity. Hence, among the species tested in our study, familial affinity was not a reliable indication of species digestibility. As a whole, the family Marattiaceae performed more poorly than any other family tested (Fig 1), despite *Angiopteris evecta* being the most digestible fern species overall (Tables 2 and 3) and the third-most digestible species in the fall data set after *Equisetum hyemale* and *E. giganteum*. *Osmunda regalis*, at its peak in the spring, is similarly highly digestible, though it experiences a large reduction in gas production in the fall. The older leaves of *Marattia attenuata* are consistently the most poorly performing foliage (Fig 2A and 2B and Tables 2 and 3).

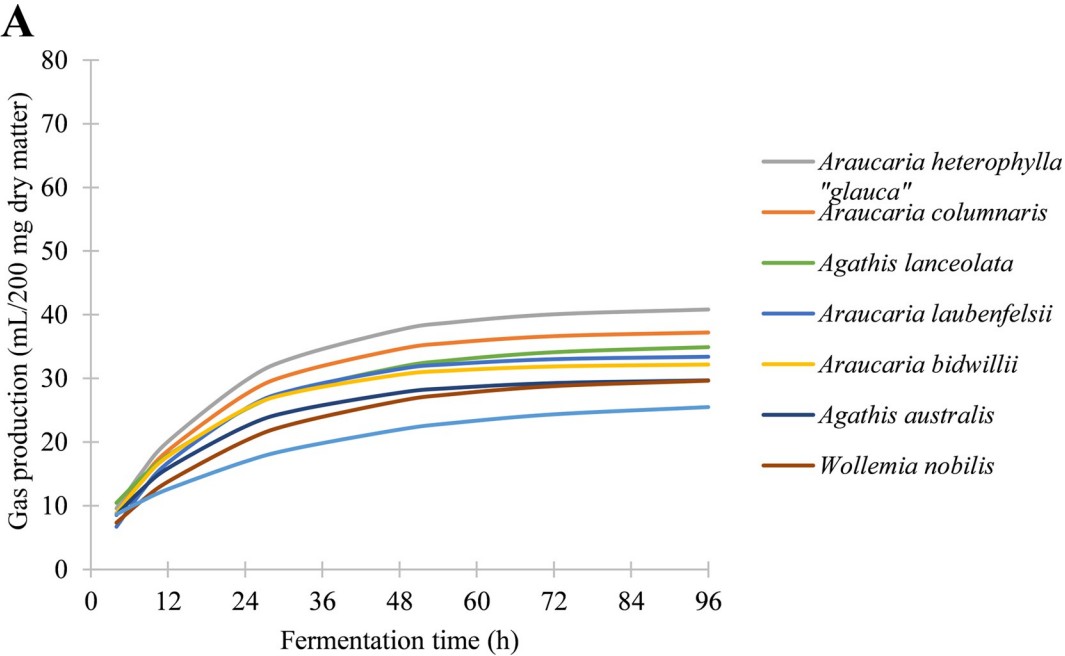

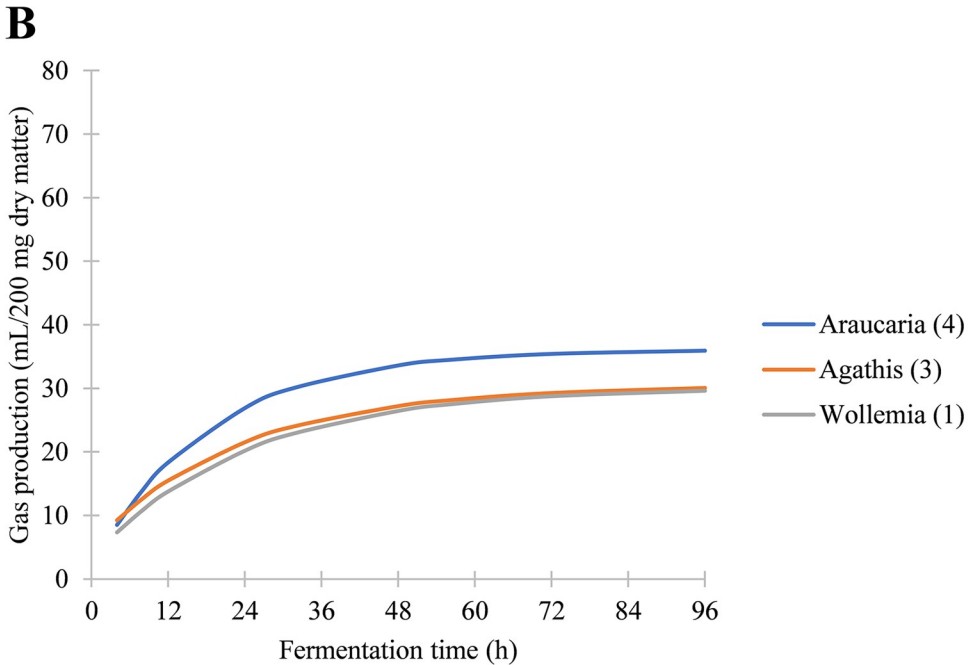

**Fig 6. Gas production of the Araucariaceae.** (A) Mean gas production of araucariaceous species, plotted as individual curves. (B) Mean gas production of araucariaceous genera, plotted as averages of all species tested.

Thus, individual fern species may have been extremely nutritious forage for herbivorous dinosaurs, but this would have been greatly influenced by environmental conditions. While a broader survey of ferns could offer deeper insights into digestibility, the results produced in this study show the wide range of energy yield that ferns may have been capable of offering. Massive, bulk-feeding megaherbivores like sauropods likely would have occasionally exploited even poorly fermenting species like *M. attenuata*, as larger animals are typically more tolerant

of poor-quality diets [27–30], although whether this is the result of an actual increase in digestive efficiency due to their large size or a result of necessity has been debated [29–31]. However, it should also be noted that the ecological preference of today's matoniaceous ferns for closed forests and the extremely slow growing nature of *Angiopteris* may have made it less likely that the Matoniaceae were frequently sought out as an abundant and readily accessible food source by large, fully grown sauropods [26].

## Digestibility of the Araucariaceae

As whole, the Araucariaceae are again confirmed to be among the most energetically nutritious taxa, particularly those of the genus *Araucaria*. The evergreen foliage of the Araucariaceae may have provided a consistent source of highly caloric food to dinosaurs that could reach the leaves of tall, mature trees, regardless of local environmental conditions. In fact, only one araucarian species, *Araucaria bidwillii*, was significantly affected by seasonality. Although the araucariaceous leaves are not as immediately digestible as *Equisetum*, the Araucariaceae, which grew in both the Northern and Southern Hemispheres during the Jurassic [26], would have provided a substantial amount of energy given four or more days for fermentation, if their nutritional qualities were similar to those in the present-day world. Large herbivores such as sauropods likely would have been capable of exploiting the fermentative behavior of araucariaceous species. Hummel et al. [1] found that the Araucariaceae were not only more calorically nutritious than the nearest living relatives of other Mesozoic plants, including cycads, podocarps, and tree ferns, but were also, over time, more digestible than the dicot leaves and grasses on which most modern herbivores feed. The results of our study confirm the high nutritional yield of these conifers, particularly those of the genus *Araucaria* (Fig 6A and 6B). With the inclusion of *Wollemia*, all three genera of the Araucariaceae have now been analyzed. While the results indicate that *Wollemia* is not as digestible as those of *Araucaria*, it is comparable in its fermentative yield and kinetic behavior to *Agathis australis* (Fig 6A).

*Araucaria heterophylla* was the most digestible member of the Araucariaceae tested in our study, with a maximum gas production of 42.3 ml/200 mg DM in the spring and 40.0 ml/200 mg DM in the fall (Tables 2 and 3). This is comparable to the maximum gas production of 41.9 ml/200 mg DM found in samples collected in 2007 (S1 Fig and S1 Table). Here, *Araucaria bidwillii* reached a maximum gas production between 27.7 and 37.4 ml/200 mg DM (Tables 2 and 3), whereas the samples collected from Australia had a maximum gas production as low as 24.8 ml/200 mg DM in the mature specimens, but as high as 29.1 ml/200mg DM in saplings (S1 Table) These results highlight the relatively uniform yield that these taxa can have, even between different individuals in variable climates, localities, and collection periods. Overall, the results of the current study confirm the high nutritional yield of these conifers, particularly those of the genus *Araucaria* (Fig 6A and 6B), though *Wollemia* and *Agathis* still provide more energy than other nonangiospermous taxa.

## Seasonal effects on digestibility

Overall, four of the 16 plants tested here were found to be significantly affected by seasonality, which encompasses factors such as temperature, the amount of sunlight, day length, and precipitation received during the growing season. Of those that experienced a significant fluctuation in digestibility between the two seasons, all taxa but the younger leaves of *Cyathea cooperi* decreased production from spring to fall (Fig 3). The species that were affected were not related; one pertained to the conifer family Araucariaceae and one each to the fern families Osmundaceae, Marattiaceae, and Cyatheaceae. In both seasons, *Equisetum hyemale* and *E. giganteum* were the most highly digestible, while the older leaves of *M. attenuata* were the least digestible

of all material tested. However, the relative digestibility of all other species varied between the two collection periods. For example, in spring, the most digestible fern was *Osmunda regalis*, but the more digestible fern in the fall collection was *Angiopteris evecta* (Fig 2).

## Leaf maturity effects on digestibility

Another potential factor for fluctuations in digestibility is leaf maturity. Consistent with studies on the digestibility of angiosperm livestock feed plants, younger leaves are generally more nutritious [32–34]. Thus, leaf maturity may influence digestibility in ferns, as is shown in the species *Marattia attenuata* (Fig 5). However, maturity effects for *Cyathea cooperi* were found to be insignificant (Fig 5). Interestingly, maturity may interact with seasonality, as the younger leaves of *M. attenuata* were not affected by seasonality, although the older leaves were (Fig 3). This combination could result in a large variation in digestibility within this species, as the more mature leaves in the fall produced only half as much gas as the younger leaves did in the spring. Such a nutritional gradient within an individual, poorly digestible plant would favor herbivores that could be more selective about which parts of the plant they consume, allowing them to target the younger, more nutritious leaves and avoiding the largely indigestible mature leaves, particularly late in the growing season. Such herbivores would include younger or smaller dinosaurs that do not need to take in as much fodder each day [31]. A larger survey could be conducted to determine the broader relevance of leaf maturity and the degree to which it interacts with seasonality within these nonangiospermous species, although the inconsistency of the results here may indicate that further experiments will not yield applicable results.

## Donor species effects on digestibility

The donor species of rumen fluid was not found to affect gas production. Two donor species—domestic sheep and cattle—were used in this experiment, and all donors followed diets outlined for use in the Hohenheim gas test [9]. These results are consistent with those of Cone et al. [35], who showed that gas production using cow rumen fluid and sheep rumen fluid were comparable after at least 48 hours of production. This supports the concept that the species of the donor does not influence digestive capability; rather, it is the diet of the animal that most strongly influences the composition of the gut biota and therefore efficiency in digestion [21]. As such, Hummel and Clauss [28] hypothesize that an animal that is accustomed to feeding on the plants analyzed here would most likely have gut flora that is even more efficient at fermenting such material than the gut flora taken from animal feeding on standardized angiosperm foliage.

## Environmental considerations

Environmental factors, such as precipitation and sunlight related to seasonality, can drastically affect the nutritional yield of plants [36]. In our study, the site of collection at the Huntington Botanical Gardens in San Marino, California, is classified as having a mild temperate climate with warm, dry summers [37]. Climate data from the nearby city of Pasadena document a maximum temperature of 34.4˚C and a minimum temperature of 11.1˚C in the period between the two collection dates in 2018. Precipitation was low, as 12.8 mm of precipitation was recorded for May and none for the period from June to September [38].

In comparison, the climate during the Jurassic was quite different. According to Sellwood and Valdes [39], the Jurassic climate was warm and equable enough that forests grew near both poles, which experienced warm summers but cold, dark, and potentially wet winters. Equatorial regions would have been more even in climate all year-round. In contrast to today's polar to tropical climate gradient, Cretaceous temperatures were up to 10˚C warmer across the

globe at its peak in the Cenomanian; warm-climate plants grew in the polar regions, although there is evidence of polar cooling by the Late Cretaceous [40].

Growth rings in Jurassic wood provide direct fossil evidence of Mesozoic plants responding to environmental conditions, along with providing paleoclimatological evidence for moderate seasonality in some localities. Such growth rings have been documented in the Triassic of China [41]; the Jurassic of India [42], England [43, 44], USA [45], and Mongolia [46]; and the Cretaceous of India [42], Brazil [47, 48], and Alaska [49]. In some cases, there is evidence that the trees were subjected to stress or trauma that interrupted growth for unpredictable or irregular periods of time, such as volcanism or localized drought, rather than climatic periodicity [50–52]. Interruptions and changes in the growth in response to these environmental incidents likely would have been reflected in the nutritional capacity of the edible plant parts, at least for short periods of time.

## Herbivore physiology and behavior

Although sauropod dinosaurs were not ruminants like the donor species used in this experiment, which have a highly derived digestive system consisting of multiple stomachs to facilitate several oral processing and digestive phases, and it is unlikely that they would have been foregut fermenters [28], the *in vitro* nature of the Hohenheim gas test minimizes the involvement of the unique digestive anatomy of ruminants while still simulating the fermentative processes that would have occurred within the gut. Because of the rarity of soft tissue evidence in dinosaur fossils, there is no direct evidence as to the nature of sauropod digestive systems; however, Hummel and Clauss [28] suggest that hindgut fermentation is the most likely form of plant digestion in sauropod dinosaurs because a foregut fermenting system is complex to evolve and would not support a high intake diet nor the lack of chewing [28]. Here, the Hohenheim gas test uses foregut-fermenting ruminants as the source of digestive microbes, but it has been found that microbial populations and their functions are highly similar in most animal species, which allows for ruminant results to be applied to non-ruminant species [53]. Furthermore, data from ruminant tests can be reliably transformed to be comparable to results derived from tests using microbes from hindgut fermenters such as horses [54].

While the specific digestive anatomy of the sauropod dinosaurs has not yet been fully determined due to a lack of soft part preservation, the incredible size of these dinosaurs would have allowed for a massive fermentation chamber capable of digesting large volumes of plant material for an extended period of time. Although larger body size does not necessarily lead to increased retention times [30, 55] as once assumed [56], extended retention times would have compensated for the slow rate of digestion of plant species such as *Agathis robusta*, *Wollemia nobilis*, or *Marattia attenuata* (Fig 2A and 2B), as well as for large particle sizes [55] due to the apparent lack of mastication [28, 57], gastric mills [58], or other particle size reduction mechanisms. However, it is noteworthy that even the most slowly fermenting and weakly digestible species, *Agathis robusta*, achieved at least 56% of gas production within the first 24 hours (Fig 4), although *in vivo* fermentation and digestion would likely take longer without particle size reduction. Meanwhile, quickly fermenting plants with high digestibility similar to *Equisetum spp.* and *Angiopteris evecta* (Fig 2A and 2B) would have been especially important for small-bodied, young, or growing dinosaurs, that would have needed to consume a larger proportion of higher quality material for their body size [31]. Histological evidence also suggests that younger sauropods may have had higher metabolic rates than adult sauropods [59], which would require either more dry matter or more energy-dense food.

For adult sauropods, Hummel et al. [1] estimates the amount of consumed dry plant matter sauropods of different body masses and metabolic behaviors would have required based on the

energetic density of the plant material, suggesting that a highly digestible diet would allow for 40% less consumed plant mass than a low-quality diet. In their estimates, the number of kilograms of dry matter required varies according to different potential metabolic rates. For example, if they possessed a similar metabolic rate to reptiles (55 kJ ME/kg BW$^{0.75}$), a 70 t sauropod would need as little as 26 kg of high-energy dry matter. However, Hummel et al. [1] believe this to be unrealistic. Therefore, metabolic rates of 280 kJ ME/kg BW$^{0.75}$ (requiring 64 kg DM/day of high energy food or 106 kg DM/day of low energy food) and a mammalian rate of 550 kJ ME/kg BW$^{0.75}$ (requiring 125 kg DM/day of high energy food or 209 kg DM/day of low energy food) were also calculated [1]. Similarly, using an intermediate metabolic rate of 280 kJ ME/kg BW$^{0.75}$, Gill et al. [12] calculates that a 10.8 t *Diplodocus* would only need to need 23.8 kg of *Equisetum hyemale* per day, as opposed to 33.2 kg of the fern species *Polypodium vulgare*. However, if digestibility and the energy gained from consumption of those plants were to decrease during part of the year or during times of environmental stress, it would necessitate a greater consumed volume of material or a preference for higher-quality plants to maintain the same energy intake. Conversely, an increase in digestibility would allow for the opposite: a lower consumed volume per day or a wider variety of plants of variable nutritional capacity [31].

Elephants, a modern analog for a bulk-feeding terrestrial megaherbivore, have been found to vary their dietary composition seasonally as different forms of nutrition become more or less available; this has been shown to be true in both the savanna-dwelling African elephant [60] and the tropical Asian elephant [61]. African elephants will consume a wider variety of plants during the dry season [60], whereas Asian elephants will expand or change the habitats they occupy in order to feed on more seasonally nutritional plants [62]. Additionally, while African elephants consume a larger volume of material during the wet season than the dry season, this likely results from the lower availability of resources during the dry season, which is reflected in a corresponding decrease in the African elephants' physical condition due to nutritional stress [63]. Similar seasonal correlations have been found for other megaherbivores, including black rhinoceroses [64], greater one-horned rhinoceroses [61], giraffes [65], African buffalo [66], and even extinct giant ground sloths [67].

Among birds, which have a strong phylogenetic relationship to dinosaurs, there are few folivores because of the metabolic and behavioral requirements necessary to make such a diet viable in a small-bodied organism [68]. Those that do engage in folivory, however, display seasonal dietary behavioral changes. Ostriches, which have been previously used as an analog for sauropod dinosaurs to investigate digestive anatomy [58], have a strong preference for particular food species, but will greatly broaden their range of food sources when feeding during the dry season [69]. Hoatzin, a South American, foregut-fermenting species, modify their diets in response to seasonal changes in the availability of their preferred food sources; although their diet predominantly consists of leaves, at the end of the dry season when trees lose their foliage, they supplement their diet with flowers [70]. The speckled mousebird also compensates for seasonal food limitations by tolerating a wide variety of resources [71]. Similarly, the white-tipped plantcutter feeds on a much wider variety of species during the winter and spends up to six times longer foraging during this time because of the limited availability of evergreen species [72].

Herbivorous dinosaurs may have exhibited similar dietary behavior changes to compensate for fluctuations in the digestibility of the Mesozoic flora or other biological needs. Isotopes in the tooth enamel of *Camarasaurus* suggests these sauropods migrated seasonally, shedding light on how they were able to occupy a seasonally dry habitat where they would have otherwise experienced water and nutritional stress [73]. There may be further evidence of seasonally induced dietary plasticity in dinosaurs; for example, Chin et al. [74] interprets coprolite evidence of crustacean consumption by ornithischian dinosaurs as a seasonal feeding habit, possibly in response to nutritional needs induced by reproductive cycles, suggesting that even bulk-

feeding herbivorous dinosaurs engaged in targeted feeding when necessary. Particularly if they resulted from reproductive-stage nutritional needs, these behavioral responses to seasonal shifts in available nutrition may have been dependent on age and sex, as they are in modern herbivores. For example, Woolley et al. [75] reports that male African elephants tolerate lower-quality diets better than females do.

### Future research

Measuring other aspects of nutritional quality, such as macronutrients and C/N ratios as previously done by Midgley et al. [7], Hummel et al. [1], Wilkinson and Ruxton [8], and Gill et al. [12], may be useful in providing more insight into herbivore nutrition beyond digestibility and relative caloric value. Overall, a broader survey of plants arising from Mesozoic-age lineages would be informative, particularly of the Podocarpaceae and Cycadales. Available data on these plant groups show that the taxa that have been tested are digestively poor [1], and analyzing a broader range of species would confirm or refute these results. Additionally, further experimentation on additional living ferns with close relatives in the Mesozoic may clarify the wide range of digestive behavior seen in the Marattiaceae and Osmundaceae in our study. Testing angiosperm families with deep paleontological roots in the Cretaceous, such as water lilies (Nymphaeaceae) or palms (Arecaceae), may also provide insight into herbivory on Cretaceous angiosperms.

### Conclusions

With the inclusion of one new genus and four new species, as well as the added dimensions of seasonality and leaf maturity in nonangiospermous species, the new results reported here both support and expand upon previous conclusions regarding the high potential digestibility and energy content of the nearest living relatives of the Mesozoic flora. Most notably, all species of the horsetail genus *Equisetum* continue to be the most digestible and energetically profitable, while the conifer genus *Araucaria* is similarly confirmed to be a viable, consistently high performing food resource for herbivorous dinosaurs. Although the other Araucariaceae genera, *Agathis* and *Wollemia*, were on average slightly less digestible than *Araucaria*, gas production was still comparable to that of *Araucaria*. Ferns of the Marattiaceae and Osmundaceae are more variable between and within species, with some, such as *Angiopteris evecta*, providing consistently high nutrition, *Osmunda regalis* providing seasonably variable nutrition, and *Marattia attenuata* and *Cyathea cooperi* being more poorly digestible. Therefore, familial affinity cannot yet be used to predict the digestibility of fern species. Additionally, four of 16 samples were significantly influenced by seasonal fluctuations, and at least one was influenced by leaf maturity. However, gas production is not affected experimentally by the herbivore donor of rumen fluid. These results contribute to a more complete understanding of the nutritive capacity of Mesozoic plants for the herbivores that relied on them.

### Supporting information

**S1 Fig. Gas production of Araucariaceae spp.** Araucariaceae spp. were collected in Australia in 2007 (Gee, unpubl. data).
(TIF)

**S1 Table. Parameters for the fermentation of Araucariaceae spp.** Araucariaceae spp. were collected from naturally growing forest trees by CTG from Northern Queensland, Australia, in 2007. For each species, cumulative gas production up to 72 h and fermentative parameters in which $a + b$ is maximum gas production (ml/200 mg DM), and $c$ is rate of gas production are

given (Gee, unpubl. data).
(DOCX)

**S2 Table. A.** Means and standard deviations (SD) for all samples as a seasonal average (Combined), and for the individual fall and spring datasets. **B.** Results of Tukey post hoc tests for species-species comparisons. **C.** ANOVA results for variables influencing gas production, including species, genus, family, rumen fluid donor, season, and maturity. **D.** Post hoc Tukey test results for the significance of season on gas production. The variable is the sample in the spring compared to the same sample in the fall. **E.** Results of Tukey post hoc test for comparisons of genera within the Araucariaceae family. **F.** Tukey post hoc results comparing gas production between families. **G.** Results of Tukey post hoc tests comparing significance of maturity in the species *Cyathea cooperi* and *Marattia attenuata*. **H.** Results of Tukey post hoc test for the interaction between season and maturity in the species *Cyathea cooperi* and *Marattia attenuata*.
(ZIP)

**S3 Table. Digestion parameters resulting from the non-linear regression using PROC NLIN in SAS.** Parameters given are A, B, and c, with the estimate, standard error (SE), alpha, lower interval, upper interval, *t*-value, probability for each parameter of each sample.
(DOCX)

## Acknowledgments

We sincerely thank James P. Folsom, Director Emeritus, and Sean Lahmeyer, Plant Collections and Conservation Manager, of the Huntington Botanical Gardens for access to the living collections, as well as Kelly Shunn for assistance in collecting and preparing the specimens. We are also grateful to Brianne Palmer (University of Bonn, Germany) for helpful advice on statistical analyses. We thank Julian Andres Castillo Vargas and a second, anonymous reviewer for their helpful comments on the manuscript.

## Author Contributions

**Conceptualization:** Carole T. Gee.

**Data curation:** Christian Böttger, Karl-Heinz Südekum.

**Formal analysis:** Mariah M. Howell, Christian Böttger.

**Investigation:** Mariah M. Howell, Carole T. Gee.

**Methodology:** Carole T. Gee.

**Project administration:** Carole T. Gee, Karl-Heinz Südekum.

**Resources:** Karl-Heinz Südekum.

**Supervision:** Carole T. Gee, Christian Böttger.

**Validation:** Mariah M. Howell.

**Visualization:** Mariah M. Howell.

**Writing – original draft:** Mariah M. Howell.

**Writing – review & editing:** Mariah M. Howell, Carole T. Gee, Christian Böttger, Karl-Heinz Südekum.

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
