## [Decision Letter · Decision Letter 0]

22 Aug 2022

PONE-D-22-20691Digestibility of dinosaur food plants revisited and expanded: Previous data, new taxa, microbe donors, foliage maturity, and seasonalityPLOS ONE

Dear Dr. Howell,

Thank you for submitting your manuscript to PLOS ONE. After careful consideration, we feel that it has merit but does not fully meet PLOS ONE’s publication criteria as it currently stands. Therefore, we invite you to submit a revised version of the manuscript that addresses the points raised during the review process.

We look forward to receiving your revised manuscript.

Kind regards,

Juan J Loor

Academic Editor

PLOS ONE

Journal Requirements:

Reviewers' comments:

Reviewer's Responses to Questions

**Comments to the Author**

1. Is the manuscript technically sound, and do the data support the conclusions?

Reviewer #1: No

Reviewer #2: Partly

2. Has the statistical analysis been performed appropriately and rigorously? 

Reviewer #1: No

Reviewer #2: N/A

3. Have the authors made all data underlying the findings in their manuscript fully available?

Reviewer #1: Yes

Reviewer #2: Yes

4. Is the manuscript presented in an intelligible fashion and written in standard English?

Reviewer #1: Yes

Reviewer #2: Yes

5. Review Comments to the Author

Reviewer #1: Dear editor,

I reviewed the manuscript entitled: “Digestibility of dinosaur food plants revisited and expanded: Previous data, new taxa, microbe donors, foliage maturity, and seasonality”. Some comments are provided in the attached file for the authors guidance.

Despite the manuscript provides interesting information regarding a topic dealing with the inference of biological value of plants for dinosaurs, I found it difficult to read. This was because, in some part, it was difficult to interpret the results due to a no clear presentation of the significance for comparisons in numeric data and figures.

My first major concern focused on the authors, apparently, did not make experimental replicates, but also laboratory replicates (see comment in line 146). All the in vitro experiments need to be repeated at least in two different days. In this sense, data need to be analyzed under a mixed approach, but not a fixed one. Additionally, more details regarding experimental methods used for in vitro runs need to be provided.

A second major concern relayed in that authors did not clarify the statistical method used for comparing their results, as well as the significance for those comparisons (see comments beginning in line 161), in different parts of the manuscript. This lack of presentation of the significancy for treatment data comparison difficulted my interpretation of the result.

In general, author need to explain what the experimental replicates were used in the in vitro assays (in my opinion, these are absent) and provide the significance of their results. In several cases, they did comparisons without providing the statistical significance of those comparisons.

Reviewer #2: The paper: “Digestibility of dinosaur food plants revisited and expanded: Previous data, new taxa, microbe donors, foliage maturity, and seasonality” by Mariah M. Howell et al., is an interesting hypothetic approach to some “fanta-biology” problems; nevertheless, I have some questions and suggestions for the authors.

Anyhow, I think that this paper could be published in a Journal of basic Science and not related to animal production.

Introduction:

No matter for the general introduction contents; however, to be considered as feeds, the plant materials must for sure contain sources of nutrients in a digestible manner, but they must be also palatable and without unacceptable levels of toxic or anti-nutritional substances. I am not sure that many of the sampled plants satisfy some of these prerogatives. Therefore, the selection of plants would be on these bases and a first step in this direction could be the knowledge whether such plants are eaten by some domestic or wild still living animals.

All these aspects, as well as the suggestion in lines 479-480: “…it may also indicate that these responses are species dependent and therefore uninformative to the characterization of fossil taxa.”, would be considered before a so deep study regarding so many plants and factors affecting their nutritional traits.

Materials and Methods:

They seem appropriate and well described (but please to consider the comments to Introduction for the lack of some essential feed traits).

Results:

The results appear reasonable and well described but for me are meaningless for the reasons showed above: digestibility is not enough to characterize as a feed an unconventional material and there are no certainty about the comparison with Jurassic plants.

Discussion:

What written for results is valid for discussion too.

Furthermore, I suggest to consider that:

- It is well known that different environmental conditions, i.e. T°, humidity, light (besides CO2 concentration) can affect growth and chemical composition of plants with consequences on digestibility etc. And in dinosaur era many of them were different, suggesting to evaluate the previous effects in plant grown in similar conditions;

- In lines 428-433 there are data which need some attention because it is not clear if the intake values are dry matter and moreover they seem very small for a body weight of 10.8 t;

- Many comments, particularly for the effects of age, season etc. on digestibility are often obvious, particularly without the chemical composition (fiber);

- In any case there are too many speculations without possibility check.

Conclusions

The Authors suggest: “These results contribute to a more complete understanding of the nutritive capacity of Mesozoic plants for the herbivores that relied on them.” I can accept this, but they must “lead” me to solve my previously exposed doubts.

6. PLOS authors have the option to publish the peer review history of their article (what does this mean?). If published, this will include your full peer review and any attached files.

Reviewer #1: **Yes: **Julián Andrés Castillo Vargas

Reviewer #2: No

---

## [Author Response · Author response to Decision Letter 0]

10 Mar 2023

Reviewer #1:

I reviewed the manuscript entitled: “Digestibility of dinosaur food plants revisited and expanded: Previous data, new taxa, microbe donors, foliage maturity, and seasonality”. Some comments are provided in the attached file for the authors guidance.

Despite the manuscript provides interesting information regarding a topic dealing with the inference of biological value of plants for dinosaurs, I found it difficult to read. This was because, in some part, it was difficult to interpret the results due to a no clear presentation of the significance for comparisons in numeric data and figures.

We are grateful to Reviewer #1 for their thorough and helpful comments regarding the methodology and statistical analyses. 

My first major concern focused on the authors, apparently, did not make experimental replicates, but also laboratory replicates (see comment in line 146). All the in vitro experiments need to be repeated at least in two different days. In this sense, data need to be analyzed under a mixed approach, but not a fixed one. Additionally, more details regarding experimental methods used for in vitro runs need to be provided.

Thank you for pointing out the lack of clarity in how the procedure was presented. In our experiments, there were four laboratory replicates for each sample that were incubated in four runs in four different weeks. Two of the replicates were fermented with sheep rumen fluid, and two were incubated with steer rumen fluid. Unfortunately, experimental replicates of plant species during collection were not possible due to low number of plants (often only one) at the site of sampling. To address the reviewer’s points, we have changed the description of the replication procedure in the Materials and Methods under the Gas Production subsection to make the use of replicates clearer: 

“Four total replicates of each sample were incubated in four runs on different weeks, except for the spring Osmunda regalis (spec. no. 18-28-6) and young spring Marattia attenuata (spec. no. 18-28-7), which were each tested in two runs due to low sample volume. One sample, the fall leaves of Araucaria heterophylla “glauca” (spec. no. 18-26-33), leaked during the third run and was retested with two replicates, for a total of five complete data sets. For all other samples, two of the replicates were incubated in sheep rumen fluid, and two were incubated in steer rumen fluid. Three blanks containing rumen fluid–buffer solution without added samples, three hay standard samples, and three concentrate standard samples were included in each run to allow for calibration of results.”

A second major concern relayed in that authors did not clarify the statistical method used for comparing their results, as well as the significance for those comparisons (see comments beginning in line 161), in different parts of the manuscript. This lack of presentation of the significancy for treatment data comparison difficulted my interpretation of the result.

In general, author need to explain what the experimental replicates were used in the in vitro assays (in my opinion, these are absent) and provide the significance of their results. In several cases, they did comparisons without providing the statistical significance of those comparisons.

This is a good point. We have re-evaluated all statistical tests that we had already run and conducted new additional analyses using ANOVAs and Tukey post-hoc tests. The new statistics have now been incorporated into the manuscript, and all data are available in the supplements. The statistical methods used are also now outlined in the Materials and Methods section, including the significance level:

“The differences between net gas production between species, family, season, and maturity were calculated using ANOVAs and Tukey post-hoc tests using R version 4.2.2 [18] with a significance threshold of 0.05 (cf. [19]).”

Reviewer #2:

The paper: “Digestibility of dinosaur food plants revisited and expanded: Previous data, new taxa, microbe donors, foliage maturity, and seasonality” by Mariah M. Howell et al., is an interesting hypothetic approach to some “fanta-biology” problems; nevertheless, I have some questions and suggestions for the authors.

Anyhow, I think that this paper could be published in a Journal of basic Science and not related to animal production.

 We also thank reviewer #2 for the positive comments and thought-provoking feedback. 

Introduction:

No matter for the general introduction contents; however, to be considered as feeds, the plant materials must for sure contain sources of nutrients in a digestible manner, but they must be also palatable and without unacceptable levels of toxic or anti-nutritional substances. I am not sure that many of the sampled plants satisfy some of these prerogatives. Therefore, the selection of plants would be on these bases and a first step in this direction could be the knowledge whether such plants are eaten by some domestic or wild still living animals.

This is a good point. The toxicity and anti-nutritional aspects of these plants are very important. These points have been discussed by Hummel et al. (2008), and, in particular, Gee (2011). The latter study goes into detail about the palatability, toxicity, and availability of these taxa, both extant and fossil, as well as their consumption by extant animals. Our study follows up on these first two studies to confirm, complete, and expand the overall data set. In other words, we analyze plants that Jürgen Hummel and Carole Gee did not test in their original data set, but wish they had.

To discuss issues of palatability, toxicity, and anti-nutritional substances, we have now included a reference to the work of Gee (2011) in a brief overview in the introduction:

“…Gee [13] conducted a comprehensive review of the Mesozoic flora which ranked each taxon according to the likelihood they would be a preferred food source for herbivorous dinosaurs. This ranking was based on their availability, biomass, growth habits, recovery potential, and palatability, including gas production, toxicity, and whether modern animals are known to consume them. Based on these factors, the Equistaceae, Araucariaceae and Cheirolepidiaceae were the most likely fodder for Jurassic herbivores.” 

All these aspects, as well as the suggestion in lines 479-480: “…it may also indicate that these responses are species dependent and therefore uninformative to the characterization of fossil taxa.”, would be considered before a so deep study regarding so many plants and factors affecting their nutritional traits.

Agreed – that is exactly what we were hoping to find out. While trends relating to taxonomic affinity were identified in the study by Hummel et al. (2008), the relevance of seasonality and leaf maturity was not previously studied. They were included here to see if they warranted further study, but since these two factors seem to be species dependent, they might not be worth exploring more. 

Materials and Methods:

They seem appropriate and well described (but please to consider the comments to Introduction for the lack of some essential feed traits).

Thanks for the positive comments!

Results:

The results appear reasonable and well described but for me are meaningless for the reasons showed above: digestibility is not enough to characterize as a feed an unconventional material and there are no certainty about the comparison with Jurassic plants.

Although we agree that it is difficult to be certain about anything in the realm of paleontology, modern analogues are often a necessity in experimental paleontology. Luckily, many of the plant species examined here (for example, horsetails), are morphologically nearly identical to their Jurassic predecessors, down to the strap-like elaters on their spores, which broadly supports the comparison. We also consider comparative digestibility between taxa, rather than absolute results, to be the most pertinent information.

Discussion:

What written for results is valid for discussion too.

Furthermore, I suggest to consider that:

- It is well known that different environmental conditions, i.e. T°, humidity, light (besides CO2 concentration) can affect growth and chemical composition of plants with consequences on digestibility etc. And in dinosaur era many of them were different, suggesting to evaluate the previous effects in plant grown in similar conditions;

These are all excellent points that complicate direct comparisons. Although not every factor you mentioned has been specifically studied, the work by Gill et al. (2019) showed that while different atmospheric conditions affected the gross digestibility of the living relatives of the Mesozoic flora, the relative gas production of species to each other was largely similar to the results observed in our study and by Hummel et al. (2008). The seasonal analyses in our study also suggest that few of the taxa are significantly impacted by seasonal changes, and even among those that were, their relative gas production compared to other species did not change much. 

- In lines 428-433 there are data which need some attention because it is not clear if the intake values are dry matter and moreover they seem very small for a body weight of 10.8 t;

Thanks for pointing out that we did not explain the units properly. The intake values do seem small, but the metabolism for sauropod dinosaurs is not thought to have been very high. To make this clearer, we have now included the estimated energy requirement from the study we cited and that the estimates were all in dry matter:

“Gill et al. [12] calculates that a 10.8 t Diplodocus with a theoretical energy requirement of 280 kJ ME/kg BW-0.75/day would only need to need 23.8 kg dry matter of Equisetum hyemale per day, as opposed to 33.2 kg dry matter of the fern species Polypodium vulgare.”

- Many comments, particularly for the effects of age, season etc. on digestibility are often obvious, particularly without the chemical composition (fiber);

While we do agree that much of the information regarding plant growth factors is self-evident, given the broad audience that this study may be interesting to (animal nutritionists, paleobotanists, and paleoecologists, but also vertebrate paleontologists), we wanted to include information that not all readers may be familiar with. 

- In any case there are too many speculations without possibility check.

Conclusions

The Authors suggest: “These results contribute to a more complete understanding of the nutritive capacity of Mesozoic plants for the herbivores that relied on them.” I can accept this, but they must “lead” me to solve my previously exposed doubts.

It is true that paleoecological research requires the use of modern analogs since the source material is no longer available, which often leads to informed inference by necessity. Deduction based on modern analogs is a standard approach in paleontology and paleoecology, and has been applied with good success for centuries in studies on ancient plants, animals, plant–animal interactions, etc. In our study, determining the comparative nutritive quality of Mesozoic plant groups for dinosaur herbivory using the Hohenheim Gas Test on the nearest living relatives of the Jurassic flora (plants that are in many cases virtually identical in morphology and anatomy, and are even considered the same on the genus level as far back as the Jurassic), is an approach that has already been successfully and independently established by a series of careful studies by several different animal nutritionists and paleobotanists (Hummel et al., 2008; Gee, 2011; Gill et al., 2019; Howell, 2021).

Comments in manuscript:

Line 17: include quantitative data in the abstract and define more clearly the objective of your research

Thanks! This information has been added.

Lines 28, 45: Improve writing

Thank you – we see where the writing could be improved and have done so. 

Line 98: Is there a specific sampling technique? Please specify the type of sampling method selected.

Yes; we have now included the sampling method by adding, “All samples were clipped from the plant using garden shears…”. We appreciate the catch. 

Line 113: Cite the procedures used for processing samples.

As suggested, the reference for the drying protocol was added:

“Specimens were dried for 48 hours at 60°C as soon as possible after collection, following the protocol established by Menke and Steingass (1989).”

Line 121: With trials are you referring to “replicates”? In in vitro studies is recommended to repeat the experiment in different days to constitute experimental replicates.

Yes, exactly. Four replicates were incubated in four runs conducted on different days. Thanks for pointing out how unclear this was. We have changed “trials” to “runs,” and the replication procedure is now explained:

“Four total replicates of each sample were incubated in four runs on different weeks, except for the spring Osmunda regalis (spec. no. 18-28-6) and young spring Marattia attenuata (spec. no. 18-28-7), which were each tested in two runs due to low sample volume. One sample, the fall leaves of Araucaria heterophylla “glauca” (spec. no. 18-26-33), leaked during the third run and was retested with two replicates, for a total of five complete data sets.”

Line 125: If possible, please provide the dataset and their descriptive statistics

The data set and all statistics have now been included as supplemental information (S3 Tables A–H; S4 Table). 

Line 127: Please provide details regarding type of buffer used. There are different types of buffer used in in vitro incubations.

The composition of the buffer solution has now been added to the Materials and Methods, under Gas Production: 

 “An nXP buffer solution (6 g NH4HCO3 + 33 g NaHCO3 per liter)…”

Line 135: Please provide the chemical composition of the buffer used for in vitro incubations.

The buffer solution composition is now referenced in the Gas Production sub-section of the Materials and Methods:

 “An nXP buffer solution (6 g NH4HCO3 + 33 g NaHCO3 per liter)…”

Line 146: Please replace as: “statistical analysis”

 Done!

Line 147: I have an important implication regarding this procedure. From the previous descriptions, authors did not make “experimental replications” but also laboratory replications. The replication component is considered as a random one. Nevertheless, authors used the proc nlin with cannot allow to use a random component in the model. Please specify if authors made experimental replications in your study. This condition is essential for results validations.

We apologize for the confusion regarding the number of replicates. For each sample, there were four replicates that were run on different weeks. Unfortunately, experimental replicates of each species was not possible due to the limited number of individuals available at the botanical garden. We have explained the replication procedure more clearly in the Materials and Methods section: 

“Four total replicates of each sample were incubated in four runs on different weeks, except for the spring Osmunda regalis (spec. no. 18-28-6) and young spring Marattia attenuata (spec. no. 18-28-7), which were each tested in two runs due to low sample volume. One sample, the fall leaves of Araucaria heterophylla “glauca” (spec. no. 18-26-33), leaked during the third run and was retested with two replicates, for a total of five complete data sets.”

Line 148: In the case of non-linear mixed models, author should use the NLMIXED procedure of SAS. Please revise this suggestion.

We considered this, but the NLMIXED procedure did not seem appropriate for the current experiment. Because a clearly defined experimental design, such as that used by Liu et al. (2002), was not possible here, the curve fitting method was applied to gas volumes for each individual syringe and then estimated parameter values for each single syringe were used to compare the different species using the tests specified. Therefore, we applied the PROC NLIN procedure instead. 

Line 155: Authors need to improve the descriptions of statistical analysis. For example, what was the significance level adopted? Statements in NLN (that in my opinion is wrong) used for parameter calculation. Method of model estimation?, etc

Following your suggestions, we have significantly strengthened the statistical analyses that were performed and included the descriptions in the Materials and Methods. All data associated with the statistical analyses and the parameter calculations are included in the supplementary material. 

Lines 161, 168, 183, 191, 195, 201, 209, 220, 233: Provide the significance to this comparison. Apply to other results in this section.

The significance of each of these statements has now been included where it was possible to make statistical inferences. Otherwise, the means and standard deviations have been included. 

Line 170: Significance was not previously defined. Please define in the materials and method section.

Significance levels have now been defined in the Materials and Methods section:

“The differences between net gas production between species, family, season, and maturity were calculated using ANOVAs and Tukey post-hoc tests using R version 4.2.2 [18] with a significance threshold of 0.05 (cf. [19]).”

Line 185: You need to define the P value for tendency

True; we have now defined the P-value thresholds in the Materials and Methods section:

“The differences between net gas production between species, family, season, and maturity were calculated using ANOVAs and Tukey post-hoc tests using R version 4.2.2 [18] with a significance threshold of 0.05 (cf. [19]).”

Line 207: This test needs to be described in the materials and method section.

For consistency and accuracy, all statistical analyses have been redone using ANOVAs and Tukey-tests. These tests are now described in the Materials and Methods. 

Line 212: Complete the sentence

The sentence has been fixed. Thanks for catching the missing word. 

Literature Cited:

Gee CT. Dietary options for the sauropod dinosaurs from an integrated botanical and paleobotanical perspective. In: Klein N, Remes K, Gee CT, Sander PM, editors. Biology of the sauropod dinosaurs: Understanding the life of giants. Indiana University Press; 2011. pp. 34–56.

Gill FL, Hummel J, Sharifi AR, Lee AP, Lomax BH. Diets of giants: The nutritional value of sauropod diet during the Mesozoic. Palaeontology. 2018;61(5): 647–58.

Howell MMH. Comparative digestibility of Mesozoic plant relatives for dinosaur herbivory: Confirmation of previous data, additional taxa, microbe donor species, foliage maturity, and seasonality. M.Sc. Thesis. University of Bonn. 2021.

Hummel J, Gee CT, Südekum KH, Sander PM, Nogge G, Clauss M. In vitro digestibility of fern and gymnosperm foliage: Implications for sauropod feeding ecology and diet selection. Proc R Soc B Biol Sci. 2008;275(1638): 1015–1021.

Liu JX, Susenbeth A, Südekum KH. In vitro gas production measurements to evaluate interactions between untreated and chemically treated rice straws, grass hay, and mulberry leaves. J Anim Sci. 2002;80(2): 517–24.

---

## [Decision Letter · Decision Letter 1]

24 Apr 2023

PONE-D-22-20691R1Digestibility of dinosaur food plants revisited and expanded: Previous data, new taxa, microbe donors, foliage maturity, and seasonalityPLOS ONE

Dear Dr. Howell,

Thank you for submitting your manuscript to PLOS ONE. After careful consideration, we feel that it has merit but does not fully meet PLOS ONE’s publication criteria as it currently stands. Therefore, we invite you to submit a revised version of the manuscript that addresses the points raised during the review process.

We look forward to receiving your revised manuscript.

Kind regards,

Juan J Loor

Academic Editor

PLOS ONE

Journal Requirements:

Reviewers' comments:

Reviewer's Responses to Questions

**Comments to the Author**

1. If the authors have adequately addressed your comments raised in a previous round of review and you feel that this manuscript is now acceptable for publication, you may indicate that here to bypass the “Comments to the Author” section, enter your conflict of interest statement in the “Confidential to Editor” section, and submit your "Accept" recommendation.

Reviewer #1: All comments have been addressed

Reviewer #2: All comments have been addressed

2. Is the manuscript technically sound, and do the data support the conclusions?

Reviewer #1: Yes

Reviewer #2: Yes

3. Has the statistical analysis been performed appropriately and rigorously? 

Reviewer #1: Yes

Reviewer #2: Yes

4. Have the authors made all data underlying the findings in their manuscript fully available?

Reviewer #1: Yes

Reviewer #2: Yes

5. Is the manuscript presented in an intelligible fashion and written in standard English?

Reviewer #1: Yes

Reviewer #2: Yes

6. Review Comments to the Author

Reviewer #1: Thank you for the opportunity to review this manuscript. This manuscript is suitable to be published in PloS One.

Reviewer #2: The paper: “Digestibility of dinosaur food plants revisited and expanded: Previous data, new taxa, microbe donors, foliage maturity, and seasonality” by Mariah M. Howell et al., has been revised by the authors according to the reviewer suggestions (or explaining their positions). Despite some points remain arguable, I agree with the new version, but still, I have some questions and suggestions for the authors.

Introduction:

Lines 71-72, …rumen liquid… and …intestinal microflora… do not get along;

Lines 75-76, I do not understand the comment, because obviously ash content allow only to know the organic matter content;

Discussion:

Line 292, perhaps it would be better to add “available” before energy;

Line 294, do you have an idea about this possibility?

Line 303-304, the highest…maximum gas production… not correct!

Line 312, …to other the species…?

Lines 325-327, the sentence does not appear complete;

Lines 429-438, some comments are quite obvious and last sentence is not clear if it is only speculation (useless);

Lines 468-473, please to better explain these concepts;

Lines 491-494, These concepts are extremely simplified, because dry matter intake is affected both by physical and chemical factors; furthermore, I insist that 30 kg of dry matter intake for a 10.8 t animal is “nothing” (an elephant has a DM intake of 1% of its body weight: 40 kg for 4.0 t).

7. PLOS authors have the option to publish the peer review history of their article (what does this mean?). If published, this will include your full peer review and any attached files.

Reviewer #1: **Yes: **Julian Andres Castillo Vargas

Reviewer #2: No

---

## [Author Response · Author response to Decision Letter 1]

27 Jul 2023

Reviewer #1:

Thank you for the opportunity to review this manuscript. This manuscript is suitable to be published in PloS One.

We are grateful for the positive review of our manuscript and appreciate all of your comments up to this point. 

Reviewer #2:

The paper: “Digestibility of dinosaur food plants revisited and expanded: Previous data, new taxa, microbe donors, foliage maturity, and seasonality” by Mariah M. Howell et al., has been revised by the authors according to the reviewer suggestions (or explaining their positions). Despite some points remain arguable, I agree with the new version, but still, I have some questions and suggestions for the authors.

We appreciate your thoughtful comments and suggestions during the review of our paper. We agree with nearly all comments that were made, and we have added references or clarifications within the text according to your suggestions. 

Introduction:

Lines 71-72, …rumen liquid… and …intestinal microflora… do not get along;

This is a good point. We have replaced “intestinal microflora” with “ruminal microflora” in the sentence.

Lines 75-76, I do not understand the comment, because obviously ash content allow only to know the organic matter content;

You are right that this was confusing. We have added further context to explain the comparison to this study. 

“This approach contrasts to the ash test that was used by Weaver [5], which quantified the caloric value of both the digestible and indigestible fractions [1]. With the ash test, it was concluded that cycads were the best source of energy for sauropods, while Equisetum was the worst [5].”

Discussion:

Line 292, perhaps it would be better to add “available” before energy;

We agree and have added the word “available” to the sentence: 

“…the Araucariaceae consistently yield high quantities of available energy…”

Line 294, do you have an idea about this possibility?

While the digestive properties of ancient taxa cannot be estimated because of the nature of fossilization, we have included more discussion about the likelihood of the conservation of these properties based on morphological similarity:

“Although chemical information of this type is not preserved, the structural morphology of Equisetum, for example, has remained largely unchanged since the Middle Triassic [20], suggesting that the digestive properties may also have been conserved through time.”

Line 303-304, the highest…maximum gas production… not correct!

Instead of using the “maximum gas production” value here, we have changed it to refer to the 96-hour gas production value. In this case, Equisetum hyemale had the greatest 96-hour gas production of any species in both the spring and fall sets (Tables 2 and 3). 

Line 312, …to other the species…?

Thanks for catching this; we’ve now fixed the sentence to read, “Compared to the other species…” 

Lines 325-327, the sentence does not appear complete;

This sentence has been rewritten to be clearer and more concise:

“Although the high silica content of Equisetum [cf. 21] is thought to inhibit the digestibility of foliage [22], the overall high performance of this genus reiterates that it would have been a consistently nutritious option for herbivorous dinosaurs.”

Lines 429-438, some comments are quite obvious and last sentence is not clear if it is only speculation (useless);

As this paper is of interest to paleontologists that have no background in animal nutrition, we felt it was appropriate to include the information here. Often, there are questions from paleontologists about whether the results of the Hohenheim gas test can vary between individuals or species, so this comparison was included. This is a good point about the last sentence, so we have added a reference and made it more obvious that is a current hypothesis:

“As such, Hummel and Clauss [28] hypothesize that an animal that is accustomed to feeding on the plants analyzed here would most likely have gut flora that is even more efficient at fermenting such material than the gut flora taken from animal feeding on standardized angiosperm foliage.”

Lines 468-473, please to better explain these concepts;

We agree that these ideas needed further explanation and have now added more information to support them. Thanks for pointing this out.

“Although sauropod dinosaurs were not ruminants like the donor species used in this experiment, which have a highly derived digestive system consisting of multiple stomachs to facilitate several oral processing and digestive phases, and it is unlikely that they would have been foregut fermenters [28], the in vitro nature of the Hohenheim gas test minimizes the involvement of the unique digestive anatomy of ruminants while still simulating the fermentative processes that would have occurred within the gut. Because of the rarity of soft tissue evidence in dinosaur fossils, there is no direct evidence as to the nature of sauropod digestive systems; however, Hummel and Clauss [28] suggest that hindgut fermentation is the most likely form of plant digestion in sauropod dinosaurs because a foregut fermenting system is complex to evolve and would not support a high intake diet nor lack of chewing [28]. Here, the Hohenheim gas test uses foregut-fermenting ruminants as the source of digestive microbes, but it has been found that microbial populations and their functions are highly similar in most animal species, which allows for ruminant results to be applied to non-ruminant species [53].”

Lines 491-494, These concepts are extremely simplified, because dry matter intake is affected both by physical and chemical factors; furthermore, I insist that 30 kg of dry matter intake for a 10.8 t animal is “nothing” (an elephant has a DM intake of 1% of its body weight: 40 kg for 4.0 t).

In a mammalian system, these estimates would be unlikely; however, the current consensus is that adult sauropod dinosaurs likely had a metabolic rate of approximately half of a mammalian, tachymetabolic rate (280 kJ ME/kg BW0.75 rather than 550 kJ ME/kg BW0.75), which accounts for the unexpectedly low amount of dry matter cited here. However, this is a very good point to raise, and we have included the information about metabolic rates so that the estimates can be considered in an appropriate context: 

“For adult sauropods, Hummel et al. [1] estimates the amount of consumed dry plant matter sauropods of different body masses and metabolic behaviors would have required based on the energetic density of the plant material, suggesting that a highly digestible diet would allow for 40% less consumed plant mass than a low-quality diet. In their estimates, the number of kilograms of dry matter required varies according to different potential metabolic rates. For example, if they possessed a similar metabolic rate to reptiles (55 kJ ME/kg BW0.75), a 70 t sauropod would need as little as 26 kg of high-energy dry matter. However, Hummel et al. [1] believe this to be unrealistic. Therefore, metabolic rates of 280 kJ ME/kg BW0.75 (requiring 64 kg DM/day of high energy food or 106 kg DM/day of low energy food) and a mammalian rate of 550 kJ ME/kg BW0.75 (requiring 125 kg DM/day of high energy food or 209 kg DM/day of low energy food) were also calculated [1]. Similarly, using an intermediate metabolic rate of 280 kJ ME/kg BW0.75, Gill et al. [12] calculates that a 10.8 t Diplodocus would only need to need 23.8 kg of Equisetum hyemale per day, as opposed to 33.2 kg of the fern species Polypodium vulgare.”

---

## [Decision Letter · Decision Letter 2]

22 Aug 2023

Digestibility of dinosaur food plants revisited and expanded: Previous data, new taxa, microbe donors, foliage maturity, and seasonality

PONE-D-22-20691R2

Dear Dr. Howell,

We’re pleased to inform you that your manuscript has been judged scientifically suitable for publication and will be formally accepted for publication once it meets all outstanding technical requirements.

Kind regards,

Juan J Loor

Academic Editor

PLOS ONE

Additional Editor Comments (optional):

Reviewers' comments:

Reviewer's Responses to Questions

**Comments to the Author**

1. If the authors have adequately addressed your comments raised in a previous round of review and you feel that this manuscript is now acceptable for publication, you may indicate that here to bypass the “Comments to the Author” section, enter your conflict of interest statement in the “Confidential to Editor” section, and submit your "Accept" recommendation.

Reviewer #2: All comments have been addressed

2. Is the manuscript technically sound, and do the data support the conclusions?

Reviewer #2: Yes

3. Has the statistical analysis been performed appropriately and rigorously? 

Reviewer #2: Yes

4. Have the authors made all data underlying the findings in their manuscript fully available?

Reviewer #2: Yes

5. Is the manuscript presented in an intelligible fashion and written in standard English?

Reviewer #2: Yes

6. Review Comments to the Author

Reviewer #2: The text has been carefully verified and modified according to the referee suggestions, therefore I do not have any further comment.

7. PLOS authors have the option to publish the peer review history of their article (what does this mean?). If published, this will include your full peer review and any attached files.

Reviewer #2: No

---

## [Editor Report · Acceptance letter]

29 Aug 2023

PONE-D-22-20691R2 

Digestibility of dinosaur food plants revisited and expanded: Previous data, new taxa, microbe donors, foliage maturity, and seasonality 

Dear Dr. Howell:

I'm pleased to inform you that your manuscript has been deemed suitable for publication in PLOS ONE. Congratulations! Your manuscript is now with our production department. 

Kind regards, 

on behalf of

Dr. Juan J Loor 

Academic Editor

PLOS ONE